# Unmasking Trees for Tabular Data

**Calvin McCarter**
mccarter.calvin@gmail.com
**BigHat Biosciences**\*

**Reviewed on OpenReview:** https://openreview.net/forum?id=0AxbTF3Ouq

## Abstract

Despite much work on advanced deep learning and generative modeling techniques for tabular data generation and imputation, traditional methods have continued to win on imputation benchmarks. We herein present UnmaskingTrees, a simple method for tabular imputation (and generation) employing gradient-boosted decision trees which are used to incrementally unmask individual features. On a benchmark for "out-of-the-box" performance on 27 small tabular datasets, UnmaskingTrees offers leading performance on imputation; state-of-the-art performance on generation given data with missingness; and competitive performance on vanilla generation given data without missingness. To solve the conditional generation subproblem, we propose a tabular probabilistic prediction method, BaltoBot, which fits a *bal*anced *t*ree *of bo*osted *t*ree classifiers. Unlike older methods, it requires no parametric assumption on the conditional distribution, accommodating features with multimodal distributions; unlike newer diffusion methods, it offers fast sampling, closed-form density estimation, and flexible handling of discrete variables. We finally consider our two approaches as meta-algorithms, demonstrating in-context learning-based generative modeling with TabPFN.

## 1 Introduction

Given a tabular dataset, it is frequently desirable to impute any missing values within that dataset, and to generate new synthetic examples. Due to the prevalence of missingness in tabular datsets, imputation has been a long-standing task in statistics and machine learning (Little and Rubin, 2019). In particular, multiple imputation methods, which produce multiple samples from the estimated conditional distribution of missing features, have proved advantageous for downstream inferential and prediction tasks (Rubin, 1996). Multiple imputation has also been used as a subroutine for counterfactual estimation (Kreindler and Lumsden, 2016; Yoon et al., 2018b) and domain adaptation (Ragab et al., 2023; McCarter, 2024). Synthetic data generation for tabular data has also seen recent interest, with applications in addressing data imbalance (Van Breugel et al., 2021; Kim et al., 2022b) and in preserving privacy (Kotelnikov et al., 2023; Gulati and Roysdon, 2024).

Imputation and generation are closely related tasks. Multiple imputation can be seen as a form of conditional generation, where the partitioning between output variables and input variables is not known in advance. Generation is then a special case of imputation where the set of observed conditioning variables is empty. Furthermore, due to the aforementioned prevalance of missingness in tabular data, generation methods also frequently need to be able to handle missingness at training time.

In this work, we are primarily focused on generation and imputation methods for users with limited data and computing resources. On data generation, recent work (Jolicoeur-Martineau et al., 2024b) (ForestDiffusion) has shown leading results on data generation using gradient-boosted trees (Chen and Guestrin, 2016) trained on diffusion or flow-matching objectives, outperforming deep learning-based approaches, particularly on smaller datasets. However, this approach tended to struggle on tabular imputation tasks, outperformed by

---

\*Work done primarily prior to joining BigHat Biosciences. Views expressed are the author's own.

MissForest (Stekhoven and Bühlmann, 2012), an older multiple imputation approach based on random forests (Breiman, 2001).

We address this shortfall by training gradient-boosted trees to autoregressively unmask features in random order, via permutation language modeling (Yang, 2019). This autoregressive approach, which we dub UnmaskingTrees, naturally performs conditional generation (i.e. imputation): we simply fill in and condition on observed values, autoregressively generating the remaining missing values. This contrasts with tabular diffusion modeling, for which the RePaint inpainting algorithm (Lugmayr et al., 2022) is employed to mediocre effect (Jolicoeur-Martineau et al., 2024b). Because the predictor for a given feature must condition on varying subsets of the other features, the ability of gradient-boosted trees to handle missing features makes them a natural choice for autoregressive modeling. Hence, we maintain the tree-based approach of Jolicoeur-Martineau et al. (2024b), while replacing their tree-based regressors with our novel tree-based probabilistic predictors, which we turn to next.

While mean-estimating regression models are satisfactory for diffusion, for autoregression we must inject noise, and hence must estimate the entire conditional distribution of each feature. We therefore revisit the long-studied problem of (tabular) probabilistic prediction (Le et al., 2005; Meinshausen and Ridgeway, 2006). Because the conditional distribution is possibly multi-modal, parametric approaches such as XGBoostLSS (März, 2019), NGBoost (Duan et al., 2020), and PGBM (Sprangers et al., 2021) are poor choices for our setting. Meanwhile, quantization of a continuous variable can model its multi-modality, but at the cost of destroying either low-resolution or high-resolution information. A diffusion-based method, Treeffuser Beltran-Velez et al. (2024), was recently proposed to address these problems. However, as a diffusion method, it suffers from slow sampling and is unable to provide closed-form density estimates; furthermore, Treeffuser does not naturally model discrete outcomes. To address these problems, we propose BaltoBot, a *bal*anced *tree of bo*osted *t*rees. For each individual variable, we recursively divide its output space with the kernel density integral (KDI) quantizer (McCarter, 2023) into a "meta-tree" of binary classifiers, which for us are gradient-boosted trees. This allows us to efficiently generate samples and estimate densities, because each sample follows only one path from root to leaf of the meta-tree. Performing regression with hierarchical classification proved successful in computer vision object bounding box prediction (Li et al., 2020), but has been surprisingly underexplored in tabular ML and in generative modeling.

Our two methods are in fact meta-algorithms that, in combination, can create a generative model out of *any* probabilistic binary classifier. To demonstrate this flexibility, we swap out XGBoost (Chen and Guestrin, 2016) for TabPFN (Hollmann et al., 2022). TabPFN is a deep learning model pretrained to perform in-context learning for tabular classification. While it has state-of-the-art classification benchmark performance (McElfresh et al., 2024), it does not perform regression tasks, nor does it inherently perform generative modeling. Constructing a generative model out of TabPFN (Hollmann et al., 2022) was first proposed in TabPFGen (Ma et al., 2024), which approximates the posterior from TabPFN-provided likelihoods by iteratively applying stochastic gradient Langevin dynamics (Welling and Teh, 2011). But unlike the previous work, ours requires only a few TabPFN forward-passes for each sample rather than many iterative data updates.

We showcase UnmaskingTrees on two tabular case studies, and on the benchmark of 27 tabular datasets presented by Jolicoeur-Martineau et al. (2024b). On this benchmark for "out-of-the-box" performance on small tabular datasets, our approach offers leading performance on imputation and state-of-the-art performance on generation given data with missingness; and it has competitive performance on vanilla generation without missingness. We also demonstrate that BaltoBot is on its own a promising method for probabilistic prediction, showing its advantages on synthetic case studies and on a heavy-tailed sales forecasting benchmark.

Finally, we provide code with an easy-to-use sklearn-style API at `https://github.com/calvinmccarter/unmasking-trees`. In addition to being useful for practitioners, we hope our work sparks study within the tabular ML community about whether diffusion or autoregression is better for tabular data. Previous autoregressive tabular modeling methods, TabMT (Gulati and Roysdon, 2024) and DP-TBART (Castellon et al., 2023), use Transformer (Vaswani, 2017) models, making them less applicable for the GPU-poor; they also lack publicly-available implementations. Our simple, efficient implementations of UnmaskingTrees and BaltoBot contribute to investigating this question.

## 2 Method

### 2.1 UnmaskingTrees for tabular joint distribution modeling

UnmaskingTrees combines gradient-boosted trees with the training objective of generalized autoregressive language modeling (Yang, 2019), inheriting the benefits of both. Consider a dataset with $N$ samples and $D$ features. We learn the joint distribution over $D$-dimensional sample $\mathbf{x}$ by maximizing the expected log-likelihood with respect to all possible permutations of the factorization order,

$$\log p(\mathbf{x}) = \log \mathbb{E}_{\sigma \in \mathcal{U}(G_D)} \Big[ \prod_{t=1}^{D} p\big(x_{\sigma(t)} | \mathbf{x}_{\sigma(<t)}\big) \Big],$$

where $\sigma$ is a permutation drawn uniformly from $\mathcal{U}(G_D)$, the permutation group on $D$ features; $\mathbf{x}_{\sigma(<t)}$ denotes all features that precede the $t$-th feature in the permuted sequence of features. If we were to have marginalized over permutations, we would have obtained a masked language modeling procedure with a randomly-sampled masking rate $r \sim \mathcal{U}(0,1)$ (Liao et al., 2020; Kitouni et al., 2023; 2024); such a procedure was previously shown to have benefits in combination with tabular Transformer models (Gulati and Roysdon, 2024) (TabMT).

For each sample, we generate new training samples by randomly sampling an order over the features, then incrementally masking the features in that random order. Given duplication factor $K$, we repeat this process $K$ times with $K$ different random permutations, leading to a training dataset with $KND$ samples. Given this, we train XGBoost (Chen and Guestrin, 2016) models to predict each unmasked sample given the more-masked sample derived from it, one per feature. We model categorical features via softmax-based classification with cross-entropy loss; our approach for non-categorical features is described in Section 2.2.

For both generation and imputation, we generate features of each sample in random order. For imputation rather than generation tasks, we begin by filling in each sample with the observed values, and run inference on the remaining unobserved features. Implementing this is very simple: it requires about 70 lines of Python code for training, and about 20 lines for inference. The training algorithm for UnmaskingTrees is given in Algorithm 1.

---

**Algorithm 1** Unmasking Trees training

---

**Require:** dataset $\mathbf{X} \in \mathbb{R}^{N \times D}$; duplication factor $K$.
 1: {# Build self-supervised training set}
 2: Set $\mathbf{X}_{\text{train}} = \emptyset$, $\mathbf{Y}_{\text{train}} = \emptyset$.
 3: **for** $k = 1, \ldots, K$ **do**
 4:    **for** $n = 1, \ldots, N$ **do**
 5:       Draw random permutation $\sigma$ from $\mathcal{U}(G_D)$
 6:       Set $\boldsymbol{x} := \mathbf{X}_{n,:}$ and $\boldsymbol{y} := \mathbf{X}_{n,:}$.
 7:       **for** $d = 1, \ldots, D$ **do**
 8:          Mask random element $\boldsymbol{x}_{\sigma(d)} := [\text{MASK}]$.
 9:          Append $\mathbf{X}_{\text{train}} := \mathbf{X}_{\text{train}} \cup \{\boldsymbol{x}\}$, $\mathbf{Y}_{\text{train}} := \mathbf{Y}_{\text{train}} \cup \{\boldsymbol{y}\}$
10:       **end for**
11:    **end for**
12: **end for**
13: {# Train conditional generation models}
14: **for** $d = 1, \ldots, D$ **do**
15:    **if** feature $d$ in $\mathbf{X}$ is a categorical feature **then**
16:       Train XGBClassifier on $([\mathbf{X}_{\text{train}}]_{:,j \neq d}, [\mathbf{Y}_{\text{train}}]_{:,d})$.
17:    **else**
18:       Run BaltoBot with $([\mathbf{X}_{\text{train}}]_{:,j \neq d}, [\mathbf{Y}_{\text{train}}]_{:,d})$.
19:    **end if**
20: **end for**

---

## 2.2 BaltoBot for tabular probabilistic prediction

In diffusion models, predicting the score function can be framed as a regression problem where the model learns to estimate the conditional mean. However, a key problem when switching to autoregressively generating continuous data is that this regression approach will attempt to predict the mean of a conditional distribution, whereas we would like the model to sample from the possibly-multimodal conditional distribution. The simplest solution is to quantize continuous features into bins, because classification over histograms is inherently multimodal; TabMT (Gulati and Roysdon, 2024) did this with 1d k-Means clustering (Lloyd, 1982). Yet this not only destroys information within bins due to rounding, it also destroys information about the proximity among the ordered bins. Thus, it forces us to choose between a small number of quantization bins, yielding low resolution; or to choose a large number of bins, risking catastrophic errors due to overfitting and/or clumping of generated samples due to poor calibration. This not only limits performance, but also necessitates hyperparameter tuning (Gulati and Roysdon, 2024).

Inspired by this, we propose a general-purpose solution to the tabular probabilistic prediction problem. For each individual regression output variable, we build a height-$H$ balanced tree of binary classifiers. Consider a node with height $h$ on this "meta-tree", which is fit with $(\mathbf{X}_{\text{train}} \in \mathbb{R}^{n \times d}, \mathbf{y}_{\text{train}} \in \mathbb{R}^n)$. Using kernel density integral quantization (KDI) (McCarter, 2023), which adaptively interpolates between uniform quantization and quantile quantization, we obtain binarized $\tilde{\mathbf{y}}_{\text{train}} \in [0,1]^n$. Thus, the input space to every node is partitioned into two with the splitting point determined by KDI. We train an XGBoost classifier on $(\mathbf{X}_{\text{train}}, \tilde{\mathbf{y}}_{\text{train}})$. If $h > 0$, we then recursively pass $\{(\mathbf{X}^{(i)}, y^{(i)}) \in (\mathbf{X}_{\text{train}}, \mathbf{y}_{\text{train}}) | \tilde{y}^{(i)} = 0\}$ to its left child, and analogously for $\tilde{y}^{(i)} = 1$ to its right child. At a leaf node, $h = 0$, if given a single unique training set output value in a bin, we record this value. At inference time, given a query input $\mathbf{X}$, we descend the tree by obtaining predicted probabilities from each node's XGBoost classifier, then sampling from these. Once we reach a leaf node, we either sample uniformly from its appropriate bin, or we return the lone output value if a singleton bin.

Conceptually, our proposed approach has four advantages. First, at training and inference time, each XGBoost model within the meta-tree only sees samples that fall into its corresponding region of the output space. Thus, for a meta-tree with height $H$ (and thus $2^H$ models), each sample is only passed as input to $H$ different models. While lower-level classifiers receive less data and are poorer quality, the magnitude of such errors are smaller due to our hierarchical partitioning approach. Second, our singleton-bin technique allows us to adaptively generate discrete and mixed-type variables, if the discrete outcome is frequent relative to the total number of training samples and to the size of the meta-tree. (Up to $2^H$ discrete outcomes can be produced by BaltoBot.) Third, our adoption of KDI instead of KMeans for feature quantization is beneficial because tabular data often has features which are irregular or highly skewed. KMeans clustering is based on mixture modeling with equal-sized Gaussian components; in constrast, KDI is a 1d density-based clustering method looks for local minima after applying a smoothing transformation. KDI was previously shown (McCarter, 2023) to be better at discretizing such irregular features than KMeans, so we propose adopting it here for generative modeling. Finally, eschewing diffusion modeling enables us to perform closed-form conditional density estimation.

The training and inference algorithms for BaltoBot are given in Algorithms 2 and 3, respectively.

---

**Algorithm 2** BaltoBot training

---

**Require:** dataset $(\mathbf{X} \in \mathbb{R}^{N \times D}, \mathbf{y} \in \mathbb{R}^N)$; BaltoBot meta-tree height $H$;
1: **if** H = 0 **or** unique($\mathbf{y}$) = $C$ for some constant $C$ **then**
2:    Save bounds := $(\min(\mathbf{y}), \max(\mathbf{y}))$.
3: **else**
4:    Obtain split point $p$ from KDI quantization on $\mathbf{y}$.
5:    Train XGBoost binary classifier on $(\mathbf{X}, \mathbf{1}\{\mathbf{y} \leq p\})$.
6:    Train "left-child" BaltoBot on $\{(\mathbf{X}^{(i)}, \mathbf{y}^{(i)}) \in (\mathbf{X}, \mathbf{y}) | \mathbf{y}^{(i)} \leq p\}$, with height $H - 1$.
7:    Train "right-child" BaltoBot on $\{(\mathbf{X}^{(i)}, \mathbf{y}^{(i)}) \in (\mathbf{X}, \mathbf{y}) | \mathbf{y}^{(i)} > p\}$, with height $H - 1$.
8: **end if**

---

---
**Algorithm 3** BaltoBot inference
---
**Require:** input query $\mathbf{x} \in \mathbb{R}^D$; trained BaltoBot model.
 1: **if** bounds is defined **then**
 2:     Sample uniformly from $U(\texttt{bounds})$.
 3:     Return.
 4: **else**
 5:     Obtain *prediction* from XGBoost binary classifier.
 6:     **if** *prediction* = left-child **then**
 7:       Run inference on "left-child" BaltoBot with input query $\mathbf{x}$.
 8:     **else if** *prediction* = right-child **then**
 9:       Run inference on "right-child" BaltoBot with input query $\mathbf{x}$.
10:     **end if**
11: **end if**
---

### 2.3 Computational complexity

ForestDiffusion, with $T$ diffusion steps and duplication factor $K$, constructs a training dataset of size $TKN \times D$. Given the same duplication factor $K$, UnmaskingTrees will construct a training dataset of size $KND \times D$. Meanwhile, ForestDiffusion must train $DT$ different XGBoost regression models. We, on the other hand, train $D$ different BaltoBot models, one per feature; with BaltoBot meta-tree height of $H$, we then train a total of $D2^H$ XGBoost binary classifiers. However, classifiers lower in the BaltoBot meta-tree become progressively faster to train. Indeed, each constructed training sample will be seen by $DT$ different XGBoost regressors with ForestDiffusion, but only $DH$ classifiers with our approach. Given that $T \sim 50$ and $H \sim 4$, this yields a large speedup for our approach.

The KDI quantizer (McCarter, 2023) has negligible contribution to runtime, because it uses the polynomial-exponential kernel density estimator (KDE) (Hofmeyr, 2019), which has linear complexity in sample size for 1d data, unlike the quadratic complexity of the Gaussian KDE.

At inference time, each ForestDiffusion generated sample passes through $T$ steps of the diffusion reverse-process, for a total of $DT$ XGBoost predictions. For UnmaskingTrees with BaltoBot, each generated sample instead requires only $DH$ XGBoost predictions, because each sample follows only one path from root to leaf of the meta-tree. The resulting speedup is especially impactful for the multiple imputation scenario, where inference time dominates.

We observe that MissForest and MICE-Forest require training $DT$ models, where number of iterations $T$ is determined by a stopping criterion that tests for convergence. Assuming that dataset size does not affect the number of required iterations, these methods each have complexity $O(ND^2)$.

### 2.4 In-context learning-based generation with BaltoBoTabPFN and UnmaskingTabPFN

Within our flexible frameworks for joint and conditional modeling, TabPFN (Hollmann et al., 2022) can be used as a base learner for probabilistic prediction and generative modeling. For UnmaskingTabPFN joint modeling, a difficulty arises from TabPFN's inability to handle inputs $\mathbf{X}_{\text{train}}$ with missing values (NaNs). To address this, we developed NanTabPFN, a wrapper for TabPFN that supports missingness in both training and test features. Based on each test query $\mathbf{x}_{\text{test}}$, we select row indices $\mathcal{R}$ and column indices $\mathcal{C}$ so that $[\mathbf{X}_{\text{train}}]_{\mathcal{R},\mathcal{C}}$ has no NaNs, using the following key idea. Consider a particular train sample $\mathbf{x}_{\text{train}}$ and test query $\mathbf{x}_{\text{test}}$, with visible (non-missing) features denoted by sets $\mathcal{V}(\mathbf{x}_{\text{train}})$ and $\mathcal{V}(\mathbf{x}_{\text{test}})$. We can maximize the number of utilized features, while also ensuring that TabPFN receives no NaNs, by restricting the set of columns to those observed for the test query, $\mathcal{C} := \mathcal{V}(\mathbf{x}_{\text{test}})$, then choosing training samples $\mathcal{R} := \{i | \mathcal{C} \subseteq \mathcal{V}(\mathbf{x}_{\text{train}}^{(i)})\}$. In practice, our procedure is more complicated, because the above choices may result in either empty $\mathcal{C}$ or empty $\mathcal{R}$. If $\mathcal{R}$ is empty, we incrementally set random features of $\mathbf{x}_{\text{test}}$ to missing until we are able to obtain a non-empty training set. If $\mathcal{C}$ is empty, we introduce a new all-1s feature to both $\mathbf{X}_{\text{train}}$ and $\mathbf{x}_{\text{test}}$.

## 3 Results

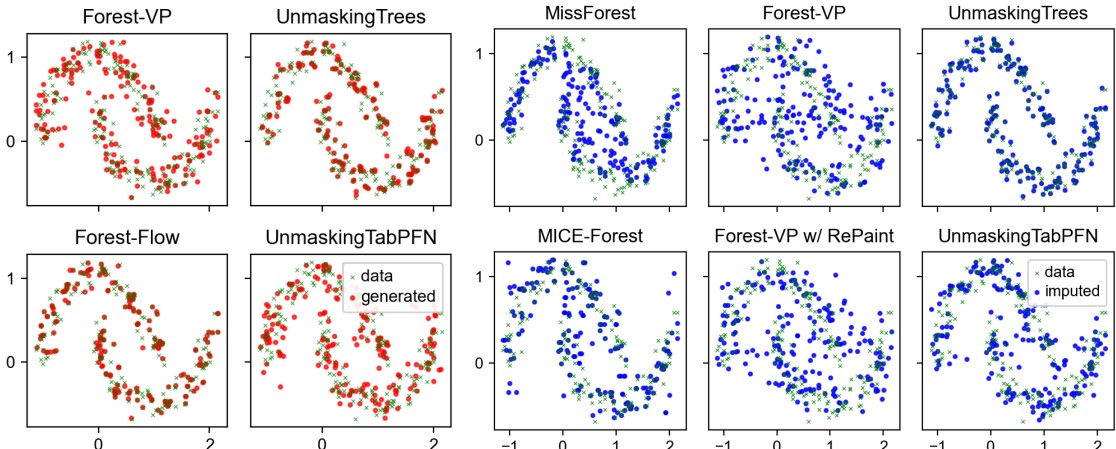

Figure 1: Results on Two Moons case study. Original data is shown in green; generated data is shown in red; imputed data is shown in blue.

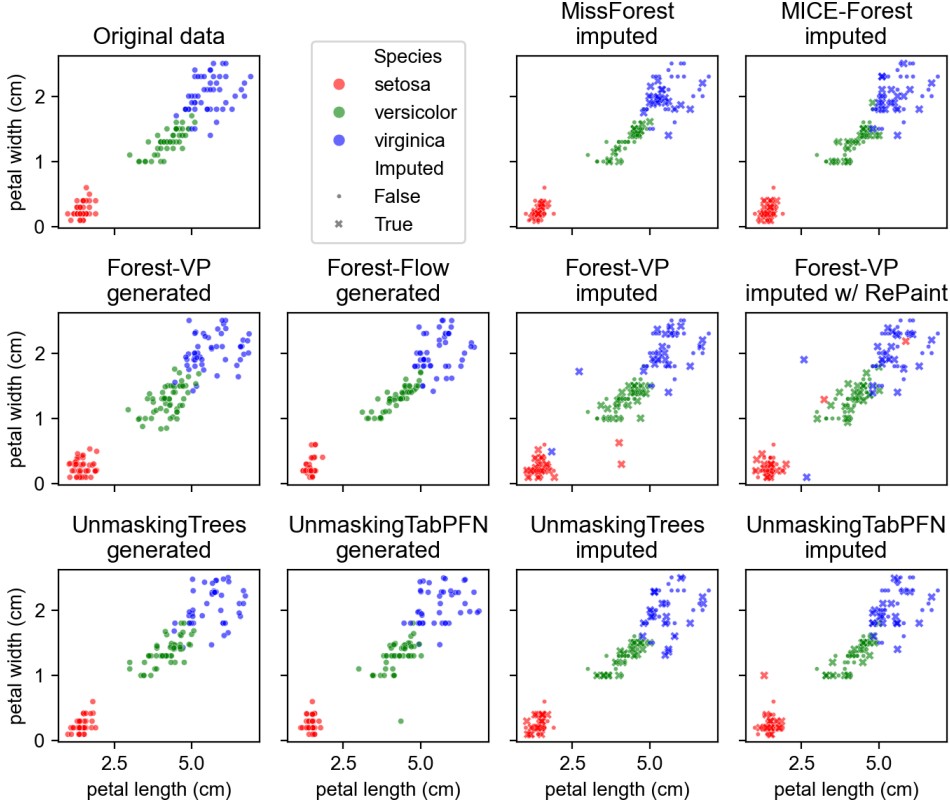

Figure 2: Results on Iris dataset, with species, petal width, and petal length depicted. Original data and synthetically-generated datasets are shown on the left columns. The imputed dataset is shown on the right columns, with × symbols highlighting the samples with any missingness that required imputation.

We evaluate UnmaskingTrees on two case studies (Section 3.1) and on a tabular benchmark of 27 datasets (Section 3.2). We then evaluate BaltoBot and BaltoBoTabPFN on tabular probabilistic prediction case studies (Section 3.3) and on a sales forecasting dataset (Section 3.4). Results were obtained always using our method's default hyperparameters: BaltoBot tree height of 4, and duplication factor $K = 50$. These hyperparameter values were tuned on the Two Moons and Iris case studies, then applied without further tuning to the remaining experiments, because tuning is problematic for users with limited computing and/or data resources. (The tree height $H$ was tuned on Two Moons and Iris by increasing $H$ until we saw that the resulting plots stopped showing visible improvement.) XGBoost hyperparameters were set to their defaults. Experiments were performed on a iMac (21.5-inch, Late 2015) with 2.8GHz Intel Core i5 processor and 16GB memory.

Overall, UnmaskingTrees (using BaltoBot) has leading performance on imputation and state-of-the-art performance on generation after training on incomplete data; and it has competitive performance on vanilla tabular generation scenarios. We further demonstrate the benefits of BaltoBot and BaltoBoTabPFN when evaluated in their own right for probabilistic prediction.

### 3.1 Case studies on Two Moons and Iris datasets

**Two Moons dataset** We first compare our approach to previous leading methods on the synthetic Two Moons dataset with 200 training samples and noise level $\mathcal{N}(0, 0.1)$. We compare UnmaskingTrees to MissForest (Stekhoven and Bühlmann, 2012), MICE-Forest (Van Buuren et al., 1999; Wilson et al., 2022) (another popular traditional multiple imputation method), and ForestDiffusion, with default hyperparameters for all methods. For ForestDiffusion, we evaluate both the variance-preserving SDE diffusion (Forest-VP) and flow-matching (Forest-Flow) versions on generation; on imputation, we evaluate Forest-VP with and without RePaint, again using default RePaint hyperparameters; Forest-Flow does not support imputation.

We show results in Figure 1. On generation, Forest-VP appears to do best according to visual inspection, while UnmaskingTrees and Forest-Flow perform similarly decently. UnmaskingTabPFN performs poorly, but does capture the overall shape of the distribution. Next, we turn to imputation, wherein we request a single imputation for a copy of the original training data with the second dimension ($y$-axis) values masked out. ForestDiffusion struggles with and without RePaint, with substantial out-of-distribution imputations, and MissForest and MICE-Forest share this problem to lesser degrees. Meanwhile, UnmaskingTrees generates impeccable imputations.

**Iris dataset** In Figure 2, we show results for the Iris dataset (Fisher, 1936), plotting petal length, petal width, and species. We compare both methods on generation, and to compare on imputation, we create another version of the Iris dataset, with missingness completely at random: we randomly select samples with 50% chance to have any missingness, and on these samples, we mask the non-species feature values with 50% chance. Visually, ForestDiffusion and UnmaskingTrees perform about equally well on generation. Meanwhile, on imputation, UnmaskingTrees does a better job conditioning on species information than ForestDiffusion. UnmaskingTrees also produces more diverse imputations than MissForest.

### 3.2 Benchmarking UnmaskingTrees on 27 tabular datasets

**Imputation** Here, we add UnmaskingTrees to the benchmark of 8 imputation methods on 27 public datasets, evaluated according to 9 metrics, developed by Jolicoeur-Martineau et al. (2024b) for evaluating tabular imputation and generation methods. This benchmark primarily contains smaller-sized (with $103 \leq N \leq 20{,}640$ and $4 \leq D \leq 90$) datasets, which our approach is especially geared towards. Namely, we compare our approach against Forest-VP Jolicoeur-Martineau et al. (2024b), as well as k-NN imputation (Troyanskaya et al., 2001), ICE (Buck, 1960), MICE-Forest (Van Buuren et al., 1999; Wilson et al., 2022), MissForest (Stekhoven and Bühlmann, 2012), Softimpute (Hastie et al., 2015), minibatch Sinkhorn optimal transport (Muzellec et al., 2020), and generative adversarial nets (GAIN) (Yoon et al., 2018a). [1] We follow Jolicoeur-Martineau et al. (2024b) in computing the per-dataset rank of each method relative to other methods, then reporting the

---

[1]We do not add TabMT (Gulati and Roysdon, 2024) and TabPFGen (Ma et al., 2024) to the benchmark because no code was provided. We do not add UnmaskingTabPFN because of out-of-memory errors on our machine.

average over 27 datasets. For all methods other than our own, we compute ranks by reusing the raw scores provided in Jolicoeur-Martineau et al. (2024b)'s code repository.

Results for imputation are shown in Table 1. UnmaskingTrees wins first place on 3/9 metrics, including both metrics based on downstream prediction tasks; and it generally outperforms ForestDiffusion, winning on 8/9 metrics. While MissForest wins first place on 4/9 metrics, UnmaskingTrees wins 5-4 head-to-head vs MissForest; UnmaskingTrees has average *averaged rank* of 3.2 compared to 3.5 for MissForest. UnmaskingTrees is also the only method with better than 5th place rank on all metrics.

We report further ablation experiments in Table 2, wherein we run UnmaskingTrees without BaltoBot, and instead with vanilla quantization using k-Means clustering (Lloyd, 1982) and KDI quantization (McCarter, 2023). Results showing progressive improvements for the UnmaskingTrees framework, for KDI quantization versus k-Means, and for the BaltoBot method used in our full proposed solution.

Table 1: Tabular data imputation (27 datasets, 3 experiments per dataset, 10 imputations per experiment) with 20% missing. Shown are *averaged rank* over all datasets and experiments (standard-error). Overall best is highlighted; better of Forest-VP versus ours is **boldface blue**.

| | MinMAE ↓ | AvgMAE ↓ | $W_{train}$ ↓ | $W_{test}$ ↓ | MAD ↓ | $R^2$ ↓ | $F_1$ ↓ | $P_{bias}$ ↓ | $Cov_{rate}$ ↓ |
|---|---|---|---|---|---|---|---|---|---|
| KNN | 5.5 (0.5) | 6.3 (0.4) | 4.9 (0.4) | 5.0 (0.4) | 8.4 (0) | 6.5 (1) | 5.7 (1.1) | 6.2 (1) | 5.4 (0.6) |
| ICE | 6.8 (0.4) | 4.7 (0.4) | 7.0 (0.5) | 7.2 (0.4) | **1.6** (0.2) | 6.2 (1) | 7.0 (0.6) | 5.7 (0.9) | 5.3 (0.6) |
| MICE-Forest | 3.9 (0.4) | **2.5** (0.2) | 2.9 (0.2) | 3.0 (0.2) | 3.6 (0.2) | 3.7 (1.4) | 3.2 (1) | 5.5 (1.2) | 4.3 (0.6) |
| MissForest | **2.7** (0.5) | 4.0 (0.4) | **1.8** (0.3) | 2.0 (0.3) | 5.5 (0.2) | 3.8 (1.4) | 2.5 (0.5) | 5.5 (1.5) | **3.3** (0.5) |
| Softimpute | 6.7 (0.4) | 7.6 (0.4) | 7.1 (0.5) | 7.3 (0.5) | 8.4 (0) | 6.0 (0.9) | 7.8 (0.4) | 6.3 (0.9) | 6.7 (0.4) |
| OT | 5.9 (0.4) | 6.1 (0.3) | 6.0 (0.5) | 6.0 (0.5) | 3.7 (0.3) | 6.2 (0.5) | 6.8 (0.6) | 5.5 (0.8) | 4.8 (0.5) |
| GAIN | 4.7 (0.4) | 6.5 (0.3) | 6.0 (0.3) | 6.0 (0.2) | 6.9 (0.1) | 5.7 (0.8) | 5.4 (0.8) | 4.7 (1) | 5.0 (0.6) |
| Forest-VP | 5.3 (0.4) | 4.0 (0.5) | 5.8 (0.3) | 5.1 (0.4) | **3.2** (0.4) | 4.5 (0.9) | 4.6 (0.8) | **3.3** (0.6) | 5.5 (0.7) |
| UTrees | **3.5** (0.5) | **3.2** (0.5) | **3.5** (0.4) | **3.5** (0.5) | 3.8 (0.2) | **2.5** (0.6) | **2.2** (0.6) | **2.3** (0.9) | **4.7** (0.6) |

Table 2: Averaged ranks from ablation study of tabular data imputation (27 datasets, 3 experiments per dataset, 10 imputations per experiment) with 20% missing. Shown are *averaged rank* over all datasets and experiments (standard-error). Overall best is highlighted; better of Forest-VP versus ours is **boldface blue**. See Table 1 for column meanings.

| | MinMAE ↓ | AvgMAE ↓ | $W_{train}$ ↓ | $W_{test}$ ↓ | MAD ↓ | $R^2$ ↓ | $F_1$ ↓ | $P_{bias}$ ↓ | $Cov_{rate}$ ↓ |
|---|---|---|---|---|---|---|---|---|---|
| KNN | 6.8 (0.6) | 7.8 (0.6) | 6.0 (0.4) | 6.1 (0.5) | 10.4 (0) | 8.2 (1.3) | 7.0 (1.5) | 7.5 (1.5) | 6.5 (0.8) |
| ICE | 8.3 (0.5) | 5.8 (0.5) | 8.5 (0.6) | 8.8 (0.5) | **1.9** (0.4) | 8.0 (1.1) | 9.0 (0.6) | 7.2 (1.1) | 6.4 (0.8) |
| MICE-Forest | 4.8 (0.6) | 3.3 (0.6) | 3.5 (0.3) | 3.4 (0.3) | 4.6 (0.4) | 4.3 (1.8) | 4.3 (1.3) | 6.8 (1.6) | 4.8 (0.7) |
| MissForest | **3.3** (0.7) | 5.0 (0.6) | **2.2** (0.4) | **2.3** (0.4) | 7.2 (0.3) | 4.7 (1.8) | 3.3 (0.9) | 6.8 (1.9) | **3.8** (0.6) |
| Softimpute | 8.3 (0.5) | 9.3 (0.5) | 8.8 (0.6) | 8.9 (0.6) | 10.4 (0) | 7.5 (1.2) | 9.8 (0.4) | 8.3 (0.9) | 7.9 (0.6) |
| OT | 7.2 (0.5) | 7.6 (0.4) | 7.4 (0.6) | 7.4 (0.6) | 4.8 (0.4) | 8.2 (0.5) | 8.8 (0.6) | 7.3 (0.7) | 5.8 (0.7) |
| GAIN | 5.8 (0.5) | 8.3 (0.4) | 7.2 (0.5) | 7.5 (0.4) | 8.9 (0.1) | 7.5 (0.8) | 7.4 (0.8) | 6.7 (1) | 6.1 (0.8) |
| Forest-VP | 6.4 (0.5) | 4.8 (0.6) | 7.0 (0.4) | 6.1 (0.5) | **3.8** (0.5) | 6.5 (0.9) | 6.6 (0.8) | 4.5 (0.8) | 6.5 (0.8) |
| UTrees-kMeans | 6.0 (0.6) | 5.8 (0.5) | 6.3 (0.6) | 6.1 (0.6) | 4.1 (0.3) | 4.0 (0.7) | **2.9** (0.6) | 3.8 (1) | 6.0 (0.7) |
| UTrees-KDI | 5.1 (0.5) | 5.1 (0.5) | 5.4 (0.6) | 5.6 (0.5) | 4.8 (0.3) | 4.5 (0.9) | 4.0 (0.5) | **3.5** (1.2) | 6.4 (0.7) |
| UTrees | **3.8** (0.5) | **3.2** (0.5) | **3.8** (0.4) | **3.8** (0.5) | 5.0 (0.3) | **2.7** (0.6) | **2.9** (0.8) | **3.5** (0.8) | **5.8** (0.7) |

**Generation with and without missingness** We next repeat the experimental setup of Jolicoeur-Martineau et al. (2024b) for evaluating tabular generation methods. For tabular generation, using the same 27 datasets, Jolicoeur-Martineau et al. (2024b) benchmark their methods (Forest-VP and Forest-Flow) against 6 other methods, namely, Gaussian Copula (Joe, 2014), tabular variational autoencoding (TVAE) (Xu et al., 2019), two conditional generative adversarial net methods (CTGAN (Xu et al., 2019) and CTAB-GAN+ (Zhao et al., 2021)), and two other tabular diffusion methods (STaSy (Kim et al., 2022a) and TabDDPM (Kotelnikov et al., 2023)). These are evaluated with 9 metrics, in the vanilla fully-observed setting and in the synthetically-induced 20% missing completely at random (MCAR) setting.

Results for partially-missing data are shown in Table 3. UnmaskingTrees is first place on 5/9 metrics; head-to-head, UnmaskingTrees beats TabDDPM 5-4, and beats Forest-Flow 6-3. Results for fully-observed data are shown in Table 4. UnmaskingTrees loses head-to-head to Forest-Flow, Forest-VP, and TabDDPM, but wins against the other methods.

Raw scores, per-dataset results, and runtimes are provided in the Appendix.

Table 3: Tabular data generation with incomplete data (27 datasets, 3 experiments per dataset, 20% missing values), MissForest is used to impute missing data except in Forest-VP, Forest-Flow, and UnmaskingTrees; *averaged rank* over all datasets and experiments (standard-error). Overall best is highlighted; better of Forest-VP versus Forest-Flow versus ours is **boldface blue**.

| | $W_{train} \downarrow$ | $W_{test} \downarrow$ | $cov_{train} \downarrow$ | $cov_{test} \downarrow$ | $R^2_{fake} \downarrow$ | $F1_{fake} \downarrow$ | $F1_{disc} \downarrow$ | $P_{bias} \downarrow$ | $cov_{rate} \downarrow$ |
|---|---|---|---|---|---|---|---|---|---|
| GaussianCopula | 7.0 (0.3) | 7.1 (0.2) | 7.2 (0.3) | 7.1 (0.3) | 6.3 (0.4) | 6.6 (0.3) | 6.7 (0.4) | 5.5 (1.0) | 7.7 (0.6) |
| TVAE | 5.2 (0.3) | 4.9 (0.3) | 5.7 (0.3) | 5.8 (0.2) | 6.0 (1.0) | 5.8 (0.5) | 5.8 (0.4) | 8.0 (0.4) | 6.2 (1.0) |
| CTGAN | 8.3 (0.2) | 8.4 (0.2) | 8.4 (0.2) | 8.3 (0.2) | 8.3 (0.3) | 8.4 (0.2) | 6.5 (0.2) | 4.8 (1.2) | 7.1 (0.7) |
| CTABGAN | 6.7 (0.4) | 6.5 (0.4) | 7.1 (0.3) | 6.8 (0.3) | 7.3 (0.6) | 7.1 (0.4) | 6.6 (0.3) | 7.5 (1.0) | 6.1 (0.6) |
| Stasy | 5.9 (0.2) | 6.1 (0.3) | 5.3 (0.2) | 5.1 (0.3) | 5.8 (0.9) | 4.4 (0.4) | 5.3 (0.4) | **3.7** (0.4) | 4.6 (1.1) |
| TabDDPM | 3.0 (0.7) | 3.4 (0.7) | 2.3 (0.5) | 2.9 (0.6) | 1.7 (0.3) | 3.3 (0.6) | 3.9 (0.6) | 3.8 (1.2) | 2.0 (0.5) |
| Forest-VP | 3.7 (0.2) | 3.2 (0.3) | 3.9 (0.2) | 3.8 (0.3) | 3.2 (0.3) | **2.3** (0.3) | 4.2 (0.4) | 4.2 (0.8) | 4.5 (1.1) |
| Forest-Flow | 3.0 (0.3) | **2.6** (0.3) | 2.6 (0.3) | 2.7 (0.2) | **3.0** (0.7) | 3.7 (0.3) | 5.0 (0.5) | 3.8 (0.9) | **3.2** (0.8) |
| UTrees | **2.1** (0.2) | 2.8 (0.3) | **2.5** (0.2) | **2.5** (0.2) | 3.3 (0.8) | 3.5 (0.5) | **1.0** (0.0) | **3.7** (0.9) | 3.7 (1.0) |

Table 4: Tabular data generation with complete data (27 datasets, 3 experiments per dataset); *averaged rank* over all datasets and experiments (standard-error). Overall best is highlighted; better of Forest-VP versus Forest-Flow versus ours is **boldface blue**.

| | $W_{train} \downarrow$ | $W_{test} \downarrow$ | $cov_{train} \downarrow$ | $cov_{test} \downarrow$ | $R^2_{fake} \downarrow$ | $F1_{fake} \downarrow$ | $F1_{disc} \downarrow$ | $P_{bias} \downarrow$ | $Cov_{rate} \downarrow$ |
|---|---|---|---|---|---|---|---|---|---|
| GaussianCopula | 7.1 (0.3) | 7.2 (0.3) | 7.3 (0.3) | 7.4 (0.3) | 6.2 (0.2) | 6.4 (0.3) | 7.0 (0.4) | 6.5 (1.1) | 7.5 (0.7) |
| TVAE | 5.3 (0.2) | 5.1 (0.2) | 5.7 (0.2) | 5.7 (0.2) | 6.5 (0.7) | 6.0 (0.5) | 5.5 (0.3) | 7.3 (0.6) | 6.7 (0.6) |
| CTGAN | 8.4 (0.1) | 8.4 (0.2) | 8.3 (0.2) | 8.1 (0.2) | 8.5 (0.2) | 8.3 (0.2) | 6.7 (0.3) | 5.3 (1.1) | 7.2 (0.5) |
| CTAB-GAN+ | 6.8 (0.3) | 6.7 (0.3) | 7.2 (0.3) | 7.1 (0.3) | 6.8 (0.4) | 6.9 (0.4) | 6.9 (0.3) | 7.7 (0.8) | 6.7 (0.8) |
| STaSy | 6.1 (0.2) | 6.3 (0.2) | 5.3 (0.2) | 5.4 (0.2) | 6.0 (1.2) | 5.1 (0.3) | 6.1 (0.3) | 4.5 (0.8) | 4.2 (1.1) |
| TabDDPM | 3.0 (0.7) | 3.9 (0.6) | 2.8 (0.5) | 3.4 (0.5) | 1.2 (0.2) | 3.8 (0.6) | 3.2 (0.4) | 3.0 (0.9) | 1.4 (0.2) |
| Forest-VP | 3.2 (0.2) | 2.8 (0.2) | 3.6 (0.3) | 3.3 (0.3) | 2.8 (0.3) | **2.2** (0.3) | 4.3 (0.4) | 3.2 (0.9) | 3.5 (0.8) |
| Forest-Flow | **1.9** (0.2) | **1.5** (0.2) | **1.7** (0.2) | **1.8** (0.2) | **2.3** (0.4) | 2.4 (0.3) | 4.3 (0.4) | **2.8** (0.5) | **2.7** (0.4) |
| UTrees | 3.1 (0.1) | 3.1 (0.2) | 3.1 (0.2) | 2.8 (0.2) | 4.7 (0.3) | 3.9 (0.3) | **1.0** (0.0) | 4.7 (0.7) | 5.2 (0.9) |

### 3.3 Evaluating BaltoBot on synthetic probabilistic prediction case studies

**Wave dataset**  We compare our approach with Treeffuser (Beltran-Velez et al., 2024) on the "wave" synthetic dataset from Treeffuser (Beltran-Velez et al., 2024), which as shown in Figure 3 is nonlinear, multimodal, and heteroskedastic. On the raw probabilistic predictions in Figure 3(A), we see that BaltoBot and BaltoBoTabPFN are (by visual inspection) able to model the conditional distribution as well as Treeffuser. Yet this case study illustrates the two advantages of BaltoBot. First, in Figure 3(B) we show the runtime of the different methods: training, sampling, and total. To train on 5000 samples, Treeffuser took 1.1s and BaltoBot took 2.6s. But to generate 5000 samples, Treeffuser took 5.0s while BaltoBot took 0.72s, for $\sim 7\times$ speedup. Second, BaltoBot offers the ability to estimate a closed-form probability density function (pdf) of the predictive distribution as shown in Figure 3(C); in contrast, Treeffuser can only sample from the predictive distribution.

**Poisson-distributed count data**  We generate 500 samples of $X_i \sim \text{Unif}[0,3], Y_i \sim \text{Poisson}(\lambda = \sqrt{X_i})$, and show probabilistic predictions for $Y$ in Figure 4. Whereas Treeffuser generates a spurious negative-valued outlier and many non-integer $Y$ samples, our approach automatically models the count-type distribution of the data.

### 3.4 Sales forecasting with uncertainty

We employ the M5 sales forecasting Kaggle dataset (Makridakis and Howard, 2020) to compare BaltoBot with other probabilistic prediction methods. The dataset has five years of sales data from ten Walmart stores, and the task requires predicting the (heavy-tailed) number of units sold given a product's attributes and previous sales. We use the exact same data preparation used for Treeffuser (Beltran-Velez et al., 2024) experiments, which yields 1k products, 120k training samples, and 10k test samples. As in the Treeffuser evaluation (Beltran-Velez et al., 2024), we evaluate probabilistic predictions with the continuous ranked probability score (CRPS), and evaluate the conditional mean predictions with the mean absolute error (MAE) and root mean-squared error (RMSE).

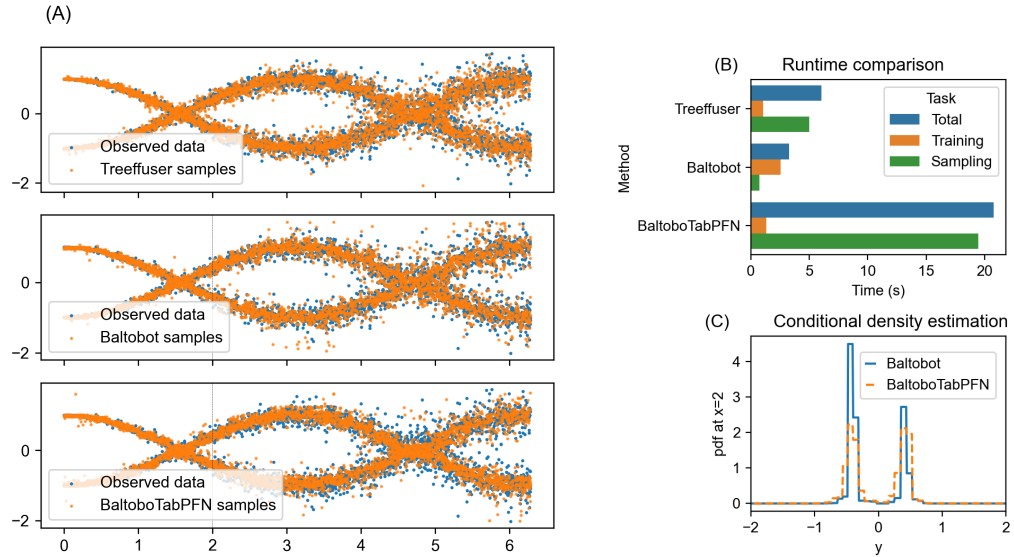

Figure 3: Comparison of Treeffuser and our approach on wave synthetic data with 5000 samples. (A) Probabilistic predictions for Treeffuser (top), BaltoBot (center), and BaltoBoTabPFN (bottom). (B) Runtime comparison for the different methods. (C) Estimated pdf from our methods at $X = 2$, depicted as the vertical dotted line in (A).

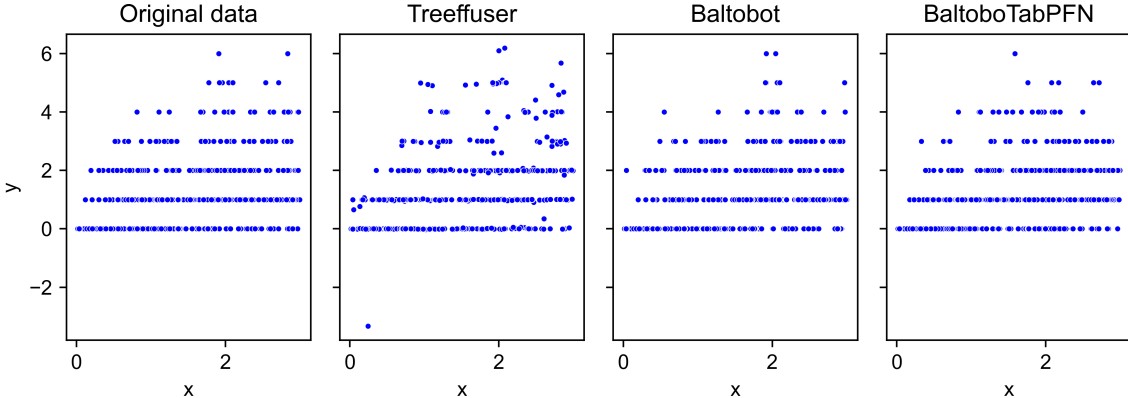

Figure 4: Comparison of Treeffuser, BaltoBot, and BaltoBoTabPFN on Poisson-distributed data. The input variable is on the x-axis, while probabilistic predictions are shown on the y-axis.

For full comparability, we follow the Treeffuser evaluation setup (Beltran-Velez et al., 2024) and evaluate CRPS by generating 100 samples from our estimators' $p(y|\mathbf{X})$ for each $\mathbf{X}$ in the testset; and for MAE and RMSE, we estimate the conditional means $\mathbb{E}[y|\mathbf{X}]$ using 50 samples. For comparability, for this (and only this) dataset, we also evaluate BaltoBot with hyperparameter tuning, using the same setup used for all other methods (10 folds, each with 80%-20% train-validation split, and 25 Bayesian optimization iterations). [2] We also compare Treeffuser, BaltoBot, and BaltoBoTabPFN when run without hyperparameter tuning.

We report results in Table 5. In addition to ours' and Treeffuser, we report results for Deep Ensembles (Lakshminarayanan et al., 2017), IBUG (Brophy and Lowd, 2022), NGBoost Poisson (Duan et al., 2020), and Quantile Regression Forests (Meinshausen and Ridgeway, 2006). For methods other than our own,

---

[2]We optimize over the following XGBoost hyperparameter spaces: `learning_rate` $\in$ log-uniform$(0.05, 0.5)$, `max_leaves` $\in \{0, 25, 50\}$, and `subsample` $\in$ log-uniform$(0.3, 1)$.

Table 5: Sales forecasting evaluation on M5 dataset. We highlight the best 2 methods for each metric. The best of Treeffuser versus ours (with tuning) is **boldface blue**; the best of Treeffuser versus ours (without tuning) is **boldface brown**.

| Method | **CRPS** $\times 10^{-1}(\downarrow)$ | **RMSE** $\times 10^{0}(\downarrow)$ | **MAE** $\times 10^{0}(\downarrow)$ |
|---|---|---|---|
| Deep Ensembles | 7.05 | 2.03 | 0.97 |
| IBUG | 8.90 | 2.12 | 1.00 |
| NGBoost Poisson | 6.86 | 2.33 | 0.99 |
| Quantile Regression Forests | 7.11 | 2.88 | 1.01 |
| Treeffuser | **6.44** | 2.09 | 0.99 |
| BaltoBot | **6.44** | **2.07** | **0.98** |
| Treeffuser (no tuning) | **6.62** | 2.09 | 0.99 |
| BaltoBot (no tuning) | 6.69 | 2.19 | 0.98 |
| BaltoBoTabPFN (no tuning) | 6.66 | **2.06** | **0.97** |

we report the metrics provided in Table 2 of (Beltran-Velez et al., 2024). Overall, our proposed methods outperform previous methods at combining excellent performance on both conditional distribution prediction and conditional mean prediction. Treeffuser and BaltoBot (both with tuning) tie for first-place according to CRPS, yet BaltoBot outperforms Treeffuser on RMSE and MAE. The winners on conditional mean metrics (RMSE and MAE) are Deep Ensembles and BaltoBoTabPFN, yet BaltoBoTabPFN (no tuning) strongly outperforms Deep Ensembles on CRPS.

## 4 Limitations and Future Work

**Limitations** While UnmaskingTrees leads *overall* on the tabular imputation benchmark, MissForest still outperformed on the metrics based on Wasserstein distance to train and test dataset distributions. And Forest-Flow still won on vanilla (i.e. no missingness) generation benchmark. It remains to be seen whether a single method can be developed which wins on all scenarios and metrics.

Our proposed BaltoBot method would benefit from a more principled method for selection of the meta-tree height $H$. Unlike with standard flat quantization where having a large number of bins can cause one to make catastrophically wrong predictions, BaltoBot "knows" proximities among meta-tree leaves. This means that making $H$ bigger tends not to cause major errors, so it's better to err on the side of larger $H$ rather than small $H$. Still, the main drawbacks with increasing $H$ are that (1) training takes longer, and (2) imputations and generations are less diverse. Deeper theoretical analysis of these trade-offs and a more principled approach for choosing the height would improve the ease-of-use and potentially the performance of our method.

While BaltoBoTabPFN performed well on probabilistic prediction tasks, when used as a subroutine in UnmaskingTabPFN, it is very slow and experienced out-of-memory errors on the (Jolicoeur-Martineau et al., 2024b) benchmark on our machine. Further improvements either to it, or to how it is employed, are needed to make it practical for all but the smallest datasets.

With regards to scalability, the runtime complexity of UnmaskingTrees training is cubic in terms of feature dimensionality $D$. This compares to quadratic complexity for ForestDiffusion; this would also be the complexity for single-order autoregressive modeling, which would support only generation rather than arbitrary imputation. This means that our approach is primarily applicable for small-sized tabular datasets, or for scenarios where inference time is more important than training time.

Related to the above remarks, we would like to emphasize that our proposed approach is aimed at and evaluated on smaller-sized tabular datasets. It is also evaluated via "out-of-the-box" performance, being aimed at users lacking the resources for large deep learning models or hyperparameter optimization. For users with access to larger tabular datasets and more extensive computing resources, recent deep learning methods like Tabsyn (Zhang et al., 2024) would be expected to perform better.

**Future Work** Reducing the training complexity with respect to the number of features is a key next step. One possibility would be to use an optimized, rather than random, selection of feature orderings at training time (Shih et al., 2022). Another possibility would be to use multi-output trees to train a single XGBoost model for all BaltoBot tree nodes and all features, similarly to a recently-proposed approach for speeding up ForestDiffusion (Cresswell and Kim, 2024). In addition to improving scalability, BaltoBot's core idea of binary partitioning could be combined with deep learning approaches. For example, one might equip a neural network with a hierarchical softmax head (Morin and Bengio, 2005) for modeling continuous outputs without losing proximity information among bins. Finally, future work could theoretically analyze BaltoBot, empirically evaluate it on uncertainty quantification and probabilistic forecasting tasks, and extend it to multiple dimensions.

## 5 Discussion and Related Work

Diffusion modeling has recently gained popularity in tabular ML (Zheng and Charoenphakdee, 2022; Jolicoeur-Martineau et al., 2024b; Beltran-Velez et al., 2024; Kotelnikov et al., 2023). Our proposed approach is an instance of the autoregressive discrete diffusion framework (Hoogeboom et al., 2021), instances of which have shown success in a variety of tasks (Yang, 2019; Austin et al., 2021; Kitouni et al., 2024; Jolicoeur-Martineau et al., 2024a). Yet our results call into question whether diffusion is beneficial for tabular conditional generation, or whether autoregression is sufficient for our setting. It has been observed that diffusion is autoregression in frequency space, progressing from low frequencies to high frequencies, which makes it a good match for image data with its power law spectra (Rissanen et al., 2022; Dieleman, 2024; Stewart, 2024). In tabular datasets without this phenomena, we would expect diffusion modeling to be less advantageous.

Why is ForestDiffusion better at vanilla generative modeling, while UnmaskingTrees is better on missing data problems? We offer two speculative explanations. First, imputation is a conditional modeling scenario, except that you do not know the partition of the features into input features and output features *a priori*. One could address imputation by learning all possible $2^D$ conditional distributions, but this is impractical for large $D$, so one would prefer to learn a single joint distribution. Both autoregression and diffusion are ways of learning a joint distribution; because autoregression does so by learning conditional distributions, it is more suited to the conditional modeling imputation setting. Second, for missing data, diffusion has a train-inference gap: during training, observed features begin the reverse process from $\mathcal{N}(0,1)$; during inference for imputation, observed features begin the reverse process at their actual values. On the other hand, the advantages of diffusion modeling (no quantization error, holistic generation, needing only an estimated score function rather than well-calibrated conditional distributions) give it superiority when these problems can be avoided.

Despite their strong outperformance on other modalities, deep learning approaches have laboured against gradient-boosted decision trees on tabular data (Shwartz-Ziv and Armon, 2022; Jolicoeur-Martineau et al., 2024b). Previous work (Breejen et al., 2024) suggests that tabular data requires an inductive prior that favors sharpness rather than smoothness, showing that TabPFN (Hollmann et al., 2022) (the leading deep learning tabular classification method) can be further improved with synthetic data generated from random forests. We anticipate that our XGBoost classifiers may be swapped out for a future variant of TabPFN that learns sharper boundaries and handles missingness.

We also note that MissForest (Stekhoven and Bühlmann, 2012), hailing from statistical literature on multiple imputation, has yet to be completely dethroned. Future progress in tabular conditional generation may require going back to the well of this traditional literature. As one example, we observe that MissForest exploits feature missingness fraction information, but we are not aware of any "machine learning" approaches which do so. The statistical literature has also previously explored the value of conditional modeling for joint modeling (Gelman and Raghunathan, 2001; Liu et al., 2014; Kropko et al., 2014). Indeed, our UnmaskingTrees approach, and all autoregressive modeling, is presaged by the full-mechanism bootstrap (Efron, 1994).

Finally, we observe where randomness enters into our generation process compared to previous work. Flow-matching (Liu et al., 2022; Albergo and Vanden-Eijnden, 2022; Lipman et al., 2022) (used in Forest-Flow) injects randomness solely at the beginning of the reverse process via Gaussian sampling, whereas diffusion modeling (Sohl-Dickstein et al., 2015; Song and Ermon, 2019) (used in Forest-VP) injects randomness both at the beginning and during the reverse process. In contrast, because our method starts with a fully-masked

sample, it injects randomness gradually during the generation process. First, we randomly generate the order over features for unmasking. Second, we do not "greedily decode" to the most likely leaf in the meta-tree, but instead sample according to predicted probabilities. Third, for continuous features, having sampled a particular meta-tree leaf bin, we sample from within the bin, treating it as a uniform distribution.

## 6 Conclusions

We proposed tree-based autoregressive modeling of tabular data, especially for data with missingness. For the subproblem of conditional probabilistic prediction of individual variables, we presented a hierarchical partitioning method with benefits over vanilla quantization and diffusion-based probabilistic prediction. We then considered each of these as meta-algorithms that enable pure in-context learning-based modeling using TabPFN as base classifier. On a benchmark for out-of-the box performance on tabular data, we showed leading results for imputation and state-of-the-art results for generation given data with missingness.

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

## A Ablation experiment with imputation – raw scores

Raw scores (shown in Table 6) demonstrate that UnmaskingTrees on its own improves upon Forest-VP's diffusion approach. We also see that KDI quantization (with 20 bins) contributes to improvement beyond k-Means (also 20 bins), and that BaltoBot yields even further improvement.

Table 6: Raw scores from ablation study for tabular data imputation (27 datasets, 3 experiments per dataset, 10 imputations per experiment) with 20% missing values. Shown are raw scores - mean (standard-error). Overall best is highlighted; better of Forest-VP versus ours is **boldface blue**. See Table 1 for column meanings.

| | MinMAE ↓ | AvgMAE ↓ | $W_{train}$ ↓ | $W_{test}$ ↓ | MAD ↑ | $R^2_{imp}$ ↑ | $F1_{imp}$ ↑ | $P_{bias}$ ↓ | $Cov_{rate}$ ↑ |
|---|---|---|---|---|---|---|---|---|---|
| KNN | 0.16 (0.03) | 0.16 (0.03) | 0.42 (0.08) | 1.89 (0.49) | 0 (0) | 0.59 (0.09) | 0.75 (0.04) | 1.27 (0.25) | 0.4 (0.11) |
| ICE | 0.1 (0.01) | 0.21 (0.03) | 0.52 (0.09) | 1.99 (0.49) | 0.69 (0.1) | 0.59 (0.09) | 0.74 (0.04) | 1.05 (0.29) | 0.39 (0.09) |
| MICE-Forest | 0.08 (0.02) | 0.13 (0.03) | 0.34 (0.07) | 1.86 (0.48) | 0.29 (0.08) | 0.61 (0.1) | 0.76 (0.04) | 0.61 (0.2) | 0.75 (0.11) |
| MissForest | 0.1 (0.03) | **0.12** (0.03) | **0.32** (0.07) | **1.85** (0.48) | 0.1 (0.03) | 0.61 (0.1) | 0.76 (0.04) | 0.62 (0.22) | **0.79** (0.08) |
| Softimpute | 0.22 (0.03) | 0.22 (0.03) | 0.53 (0.07) | 1.99 (0.48) | 0 (0) | 0.58 (0.09) | 0.74 (0.04) | 1.18 (0.34) | 0.31 (0.09) |
| OT | 0.14 (0.02) | 0.19 (0.03) | 0.56 (0.1) | 1.98 (0.49) | 0.28 (0.05) | 0.59 (0.1) | 0.75 (0.04) | 1.09 (0.27) | 0.39 (0.12) |
| GAIN | 0.16 (0.03) | 0.17 (0.03) | 0.49 (0.11) | 1.95 (0.51) | 0.01 (0.04) | 0.6 (0.1) | 0.75 (0.04) | 1.04 (0.25) | 0.54 (0.12) |
| Forest-VP | 0.14 (0.04) | 0.17 (0.03) | 0.55 (0.13) | 1.96 (0.5) | 0.25 (0.03) | **0.61** (0.1) | 0.74 (0.04) | 0.81 (0.25) | 0.57 (0.14) |
| UTrees-kMeans | 0.1 (0.02) | 0.15 (0.03) | 0.43 (0.09) | 1.9 (0.5) | **0.28** (0.06) | **0.61** (0.1) | **0.76** (0.04) | 0.63 (0.21) | **0.72** (0.13) |
| Utrees-KDI | 0.1 (0.02) | **0.14** (0.03) | 0.42 (0.09) | 1.89 (0.49) | 0.27 (0.06) | **0.61** (0.1) | **0.76** (0.04) | 0.68 (0.24) | 0.68 (0.14) |
| UTrees | 0.08 (0.02) | **0.14** (0.03) | **0.37** (0.08) | **1.87** (0.48) | 0.27 (0.07) | **0.61** (0.1) | **0.76** (0.04) | **0.55** (0.19) | 0.71 (0.13) |
| Oracle | 0 (0) | 0 (0) | 0 (0) | 1.87 (0.49) | 0 (0) | 0.64 (0.09) | 0.78 (0.04) | 0 (0) | 1 (0) |

## B Full dataset-level results

Full imputation results are in Table 7. Full generation results are in Table 8. Timing results are in Table 10, and depicted in Figure 5. Our method is relatively efficient at both imputation and generation. The datasets on which we are slowest for imputation are Libras (1976 seconds, $N = 360$, $D = 90$) and Bean (1929 seconds, $N = 13611$, $D = 16$), on our ancient 2015 iMac with 16Gb RAM. On Libras, ForestVP imputation took 12439 seconds (without RePaint) and 14715 seconds (with RePaint); on Bean, ForestVP took 898 seconds (without RePaint) and 1318 seconds (with RePaint), on their cluster of 10-20 CPUs with 64-256Gb of RAM. The datasets on which we are slowest for generation are also Libras (2987 seconds) and Bean (4346 seconds). On Libras, ForestFlow generation took 9481 seconds and ForestVP took 9042 seconds; on Bean, ForestFlow took 869 seconds and ForestVP took 947 seconds, once again on their much more powerful computing cluster.

Table 7: Full imputation results for UnmaskingTrees on (Jolicoeur-Martineau et al., 2024b) benchmark.

| Dataset | MinMAE ↓ | AvgMAE ↓ | $P_{bias}$ ↓ | $Cov_{rate}$ ↑ | $W_{train}$ ↓ | $W_{test}$ ↓ | Var ↑ | MAD (mean) ↑ | MAD (med) ↑ | $R^2$ ↑ | F1 ↑ |
|---|---|---|---|---|---|---|---|---|---|---|---|
| iris | 6.00e-02 | 8.91e-02 | 0.00e+00 | 0.00e+00 | 6.62e-02 | 2.40e-01 | 2.65e-03 | 1.48e-01 | 1.22e-01 | 0.00e+00 | 9.53e-01 |
| wine | 9.48e-02 | 1.31e-01 | 0.00e+00 | 0.00e+00 | 3.54e-01 | 1.44e+00 | 5.64e-03 | 2.37e-01 | 1.99e-01 | 0.00e+00 | 9.37e-01 |
| parkinsons | 4.69e-02 | 6.52e-02 | 0.00e+00 | 0.00e+00 | 2.94e-01 | 1.71e+00 | 2.73e-03 | 1.23e-01 | 1.03e-01 | 0.00e+00 | 8.30e-01 |
| climate | 2.38e-01 | 3.38e-01 | 0.00e+00 | 0.00e+00 | 1.22e+00 | 3.87e+00 | 4.38e-02 | 7.94e-01 | 6.88e-01 | 0.00e+00 | 7.08e-01 |
| concrete compression | 2.33e-02 | 5.19e-02 | 1.17e+02 | 2.44e-01 | 7.95e-02 | 5.05e-01 | 5.61e-03 | 1.59e-01 | 1.30e-01 | 7.55e-01 | 0.00e+00 |
| yacht hydrodynamics | 2.72e-02 | 6.53e-02 | 8.78e+01 | 9.62e-01 | 6.46e-02 | 5.11e-01 | 1.22e-02 | 1.90e-01 | 1.45e-01 | 8.96e-01 | 0.00e+00 |
| airfoil self noise | 2.42e-02 | 6.64e-02 | 3.37e+00 | 1.00e+00 | 4.60e-02 | 2.50e-01 | 1.25e-02 | 2.29e-01 | 1.84e-01 | 7.24e-01 | 0.00e+00 |
| connectionist sonar | 9.86e-02 | 1.18e-01 | 0.00e+00 | 0.00e+00 | 1.43e+00 | 8.51e+00 | 5.14e-03 | 2.22e-01 | 1.88e-01 | 0.00e+00 | 7.99e-01 |
| ionosphere | 8.52e-02 | 1.18e-01 | 0.00e+00 | 0.00e+00 | 7.82e-01 | 4.44e+00 | 1.47e-02 | 2.54e-01 | 2.02e-01 | 0.00e+00 | 9.10e-01 |
| qsar biodegradation | 1.53e-02 | 2.34e-02 | 0.00e+00 | 0.00e+00 | 1.92e-01 | 1.39e+00 | 1.25e-03 | 5.35e-02 | 4.36e-02 | 0.00e+00 | 8.49e-01 |
| seeds | 5.36e-02 | 8.45e-02 | 0.00e+00 | 0.00e+00 | 1.22e-01 | 4.78e-01 | 3.37e-03 | 1.74e-01 | 1.48e-01 | 0.00e+00 | 8.83e-01 |
| glass | 4.96e-02 | 7.59e-02 | 0.00e+00 | 0.00e+00 | 1.40e-01 | 6.42e-01 | 5.09e-03 | 1.44e-01 | 1.17e-01 | 0.00e+00 | 5.43e-01 |
| ecoli | 5.15e-02 | 8.00e-02 | 0.00e+00 | 0.00e+00 | 1.09e-01 | 4.04e-01 | 3.60e-03 | 1.54e-01 | 1.30e-01 | 0.00e+00 | 6.83e-01 |
| yeast | 4.38e-02 | 7.40e-02 | 0.00e+00 | 0.00e+00 | 1.07e-01 | 3.19e-01 | 3.58e-03 | 1.73e-01 | 1.50e-01 | 0.00e+00 | 4.44e-01 |
| libras | 3.06e-02 | 3.64e-02 | 0.00e+00 | 0.00e+00 | 6.57e-01 | 8.93e+00 | 8.11e-04 | 8.17e-02 | 7.06e-02 | 0.00e+00 | 5.69e-01 |
| planning relax | 8.41e-02 | 1.21e-01 | 0.00e+00 | 0.00e+00 | 3.06e-01 | 1.46e+00 | 5.57e-03 | 2.39e-01 | 2.03e-01 | 0.00e+00 | 4.52e-01 |
| blood transfusion | 3.44e-02 | 6.69e-02 | 0.00e+00 | 0.00e+00 | 3.21e-02 | 1.12e-01 | 4.80e-03 | 1.58e-01 | 1.32e-01 | 0.00e+00 | 5.87e-01 |
| breast cancer | 3.96e-02 | 5.16e-02 | 0.00e+00 | 0.00e+00 | 3.10e-01 | 1.85e+00 | 1.21e-03 | 1.01e-01 | 8.73e-02 | 0.00e+00 | 9.59e-01 |
| connectionist vowel | 5.30e-02 | 9.39e-02 | 0.00e+00 | 0.00e+00 | 1.88e-01 | 7.25e-01 | 5.53e-03 | 2.48e-01 | 2.14e-01 | 0.00e+00 | 6.64e-01 |
| concrete slump | 1.25e-01 | 1.88e-01 | 4.76e+01 | 7.25e-01 | 2.64e-01 | 1.16e+00 | 1.48e-02 | 3.40e-01 | 2.75e-01 | 6.75e-01 | 0.00e+00 |
| wine quality red | 4.08e-02 | 7.16e-02 | 2.01e+01 | 1.00e+00 | 1.41e-01 | 5.17e-01 | 3.52e-03 | 1.83e-01 | 1.58e-01 | 3.06e-01 | 0.00e+00 |
| wine quality white | 3.41e-02 | 6.45e-02 | 7.74e+01 | 4.78e-01 | 1.40e-01 | 4.53e-01 | 3.36e-03 | 1.91e-01 | 1.68e-01 | 3.38e-01 | 0.00e+00 |
| california | 1.97e-02 | 4.97e-02 | 2.32e+01 | 5.93e-01 | 0.00e+00 | 0.00e+00 | 4.98e-03 | 1.58e-01 | 1.40e-01 | 6.55e-01 | 0.00e+00 |
| bean | 1.02e-02 | 2.06e-02 | 0.00e+00 | 0.00e+00 | 0.00e+00 | 0.00e+00 | 1.05e-03 | 6.42e-02 | 5.80e-02 | 0.00e+00 | 7.82e-01 |
| tictactoe | 2.96e-01 | 5.11e-01 | 0.00e+00 | 0.00e+00 | 7.76e-01 | 1.93e+00 | 2.45e-02 | 1.47e+00 | 1.13e+00 | 0.00e+00 | 8.23e-01 |
| congress | 1.73e-01 | 2.77e-01 | 0.00e+00 | 0.00e+00 | 8.03e-01 | 2.38e+00 | 9.76e-03 | 5.86e-01 | 4.33e-01 | 0.00e+00 | 9.33e-01 |
| car | 4.04e-01 | 6.85e-01 | 0.00e+00 | 0.00e+00 | 4.84e-01 | 1.07e+00 | 2.95e-02 | 2.06e+00 | 1.65e+00 | 0.00e+00 | 8.01e-01 |

Table 8: Full generation results for UnmaskingTrees on (Jolicoeur-Martineau et al., 2024b) benchmark.

| Dataset | $W_{train}$ | $W_{test}$ | $cov_{train}$ | $cov_{test}$ | $R^2_{fake}$ | $F1_{fake}$ | $F1_{disc}$ | $P_{bias}$ | $Cov_{rate}$ |
|---|---|---|---|---|---|---|---|---|---|
| iris | 2.34e-01 | 3.41e-01 | 8.78e-01 | 9.16e-01 | 0.00e+00 | 9.25e-01 | 4.23e-01 | 0.00e+00 | 0.00e+00 |
| wine | 1.09e+00 | 1.53e+00 | 9.09e-01 | 9.37e-01 | 0.00e+00 | 9.15e-01 | 3.46e-01 | 0.00e+00 | 0.00e+00 |
| parkinsons | 1.34e+00 | 1.77e+00 | 7.48e-01 | 9.08e-01 | 0.00e+00 | 7.33e-01 | 3.56e-01 | 0.00e+00 | 0.00e+00 |
| climate model crashes | 3.26e+00 | 3.89e+00 | 8.96e-01 | 9.50e-01 | 0.00e+00 | 5.39e-01 | 2.81e-01 | 0.00e+00 | 0.00e+00 |
| concrete compression | 4.63e-01 | 6.21e-01 | 5.11e-01 | 8.16e-01 | 6.73e-01 | 0.00e+00 | 4.15e-01 | 1.50e+02 | 2.00e-01 |
| yacht hydrodynamics | 4.20e-01 | 6.36e-01 | 6.13e-01 | 7.89e-01 | 8.46e-01 | 0.00e+00 | 5.14e-01 | 1.42e+02 | 4.48e-01 |
| airfoil self noise | 1.93e-01 | 2.93e-01 | 6.39e-01 | 8.96e-01 | 6.09e-01 | 0.00e+00 | 4.65e-01 | 1.97e+01 | 4.78e-01 |
| connectionist bench sonar | 7.21e+00 | 8.96e+00 | 6.87e-01 | 8.89e-01 | 0.00e+00 | 7.20e-01 | 3.69e-01 | 0.00e+00 | 0.00e+00 |
| ionosphere | 3.84e+00 | 4.66e+00 | 6.11e-01 | 7.94e-01 | 0.00e+00 | 8.57e-01 | 4.22e-01 | 0.00e+00 | 0.00e+00 |
| qsar biodegradation | 1.34e+00 | 1.62e+00 | 4.81e-01 | 8.19e-01 | 0.00e+00 | 8.02e-01 | 4.44e-01 | 0.00e+00 | 0.00e+00 |
| seeds | 3.51e-01 | 5.48e-01 | 8.98e-01 | 9.63e-01 | 0.00e+00 | 8.69e-01 | 3.09e-01 | 0.00e+00 | 0.00e+00 |
| glass | 4.52e-01 | 7.12e-01 | 8.27e-01 | 9.35e-01 | 0.00e+00 | 4.41e-01 | 3.65e-01 | 0.00e+00 | 0.00e+00 |
| ecoli | 2.86e-01 | 4.38e-01 | 8.99e-01 | 9.58e-01 | 0.00e+00 | 6.16e-01 | 3.78e-01 | 0.00e+00 | 0.00e+00 |
| yeast | 2.44e-01 | 3.49e-01 | 8.54e-01 | 9.44e-01 | 0.00e+00 | 3.62e-01 | 4.32e-01 | 0.00e+00 | 0.00e+00 |
| libras | 1.01e+01 | 1.16e+01 | 4.65e-01 | 8.43e-01 | 0.00e+00 | 3.54e-01 | 3.44e-01 | 0.00e+00 | 0.00e+00 |
| planning relax | 1.02e+00 | 1.47e+00 | 9.22e-01 | 9.98e-01 | 0.00e+00 | 4.56e-01 | 3.07e-01 | 0.00e+00 | 0.00e+00 |
| blood transfusion | 1.00e-01 | 1.52e-01 | 9.62e-01 | 9.56e-01 | 0.00e+00 | 5.95e-01 | 4.07e-01 | 0.00e+00 | 0.00e+00 |
| breast cancer diagnostic | 1.55e+00 | 1.90e+00 | 7.94e-01 | 9.12e-01 | 0.00e+00 | 9.40e-01 | 3.43e-01 | 0.00e+00 | 0.00e+00 |
| connectionist bench vowel | 7.04e-01 | 8.87e-01 | 3.04e-01 | 8.34e-01 | 0.00e+00 | 5.75e-01 | 3.43e-01 | 0.00e+00 | 0.00e+00 |
| concrete slump | 6.24e-01 | 1.20e+00 | 8.71e-01 | 8.57e-01 | 5.34e-01 | 0.00e+00 | 3.52e-01 | 4.57e+01 | 5.75e-01 |
| wine quality red | 4.30e-01 | 5.40e-01 | 8.63e-01 | 9.67e-01 | 2.46e-01 | 0.00e+00 | 4.51e-01 | 5.14e+01 | 7.94e-01 |
| wine quality white | 4.23e-01 | 4.97e-01 | 8.26e-01 | 9.55e-01 | 2.52e-01 | 0.00e+00 | 4.46e-01 | 1.84e+02 | 2.83e-01 |
| california | 0.00e+00 | 0.00e+00 | 6.22e-01 | 9.03e-01 | 3.05e-01 | 0.00e+00 | 4.30e-01 | 1.75e+02 | 1.70e-01 |
| bean | 0.00e+00 | 0.00e+00 | 3.35e-01 | 7.53e-01 | 0.00e+00 | 8.16e-01 | 3.97e-01 | 0.00e+00 | 0.00e+00 |
| tictactoe | 9.44e-01 | 1.95e+00 | 8.23e-01 | 6.28e-01 | 0.00e+00 | 8.31e-01 | 2.74e-01 | 0.00e+00 | 0.00e+00 |
| congress | 1.38e+00 | 2.46e+00 | 9.11e-01 | 9.16e-01 | 0.00e+00 | 9.47e-01 | 2.87e-01 | 0.00e+00 | 0.00e+00 |
| car | 4.61e-01 | 1.05e+00 | 5.82e-01 | 5.20e-01 | 0.00e+00 | 7.99e-01 | 3.02e-01 | 0.00e+00 | 0.00e+00 |

Table 9: Full generation with 20% missingness results for UnmaskingTrees on (Jolicoeur-Martineau et al., 2024b) benchmark.

| Dataset | $W_{train}$ | $W_{test}$ | $cov_{train}$ | $cov_{test}$ | $R^2_{fake}$ | $F1_{fake}$ | $F1_{disc}$ | $P_{bias}$ | $Cov_{rate}$ |
|---|---|---|---|---|---|---|---|---|---|
| iris | 2.56e-01 | 3.54e-01 | 8.30e-01 | 8.71e-01 | 0.00e+00 | 9.42e-01 | 4.20e-01 | 0.00e+00 | 0.00e+00 |
| wine | 1.16e+00 | 1.55e+00 | 8.79e-01 | 8.96e-01 | 0.00e+00 | 9.17e-01 | 3.71e-01 | 0.00e+00 | 0.00e+00 |
| parkinsons | 1.41e+00 | 1.80e+00 | 6.49e-01 | 8.96e-01 | 0.00e+00 | 7.01e-01 | 3.76e-01 | 0.00e+00 | 0.00e+00 |
| climate model crashes | 3.33e+00 | 3.88e+00 | 8.59e-01 | 9.53e-01 | 0.00e+00 | 5.29e-01 | 3.20e-01 | 0.00e+00 | 0.00e+00 |
| concrete compression | 4.71e-01 | 6.33e-01 | 4.83e-01 | 7.89e-01 | 6.57e-01 | 0.00e+00 | 4.19e-01 | 1.64e+02 | 1.48e-01 |
| yacht hydrodynamics | 4.18e-01 | 6.25e-01 | 5.80e-01 | 8.30e-01 | 8.56e-01 | 0.00e+00 | 5.55e-01 | 1.04e+02 | 6.00e-01 |
| airfoil self noise | 2.00e-01 | 3.02e-01 | 6.05e-01 | 8.92e-01 | 5.97e-01 | 0.00e+00 | 4.56e-01 | 2.07e+01 | 3.44e-01 |
| connectionist bench sonar | 7.52e+00 | 9.11e+00 | 6.10e-01 | 8.68e-01 | 0.00e+00 | 7.30e-01 | 3.90e-01 | 0.00e+00 | 0.00e+00 |
| ionosphere | 3.99e+00 | 4.69e+00 | 5.52e-01 | 7.64e-01 | 0.00e+00 | 8.51e-01 | 4.32e-01 | 0.00e+00 | 0.00e+00 |
| qsar biodegradation | 1.37e+00 | 1.62e+00 | 4.54e-01 | 8.17e-01 | 0.00e+00 | 8.07e-01 | 4.47e-01 | 0.00e+00 | 0.00e+00 |
| seeds | 4.14e-01 | 5.86e-01 | 8.08e-01 | 9.38e-01 | 0.00e+00 | 8.53e-01 | 3.37e-01 | 0.00e+00 | 0.00e+00 |
| glass | 5.04e-01 | 7.33e-01 | 7.75e-01 | 8.93e-01 | 0.00e+00 | 4.16e-01 | 3.87e-01 | 0.00e+00 | 0.00e+00 |
| yeast | 2.51e-01 | 3.45e-01 | 8.22e-01 | 9.44e-01 | 0.00e+00 | 3.01e-01 | 4.39e-01 | 0.00e+00 | 0.00e+00 |
| libras | 1.00e+01 | 1.16e+01 | 4.60e-01 | 8.51e-01 | 0.00e+00 | 3.36e-01 | 3.63e-01 | 0.00e+00 | 0.00e+00 |
| planning relax | 1.07e+00 | 1.49e+00 | 9.01e-01 | 9.84e-01 | 0.00e+00 | 4.64e-01 | 3.37e-01 | 0.00e+00 | 0.00e+00 |
| blood transfusion | 9.11e-02 | 1.38e-01 | 9.62e-01 | 9.56e-01 | 0.00e+00 | 5.89e-01 | 4.18e-01 | 0.00e+00 | 0.00e+00 |
| breast cancer diagnostic | 1.62e+00 | 1.91e+00 | 7.48e-01 | 9.00e-01 | 0.00e+00 | 9.40e-01 | 3.69e-01 | 0.00e+00 | 0.00e+00 |
| connectionist bench vowel | 7.42e-01 | 9.15e-01 | 2.38e-01 | 8.01e-01 | 0.00e+00 | 5.46e-01 | 3.60e-01 | 0.00e+00 | 0.00e+00 |
| concrete slump | 7.16e-01 | 1.24e+00 | 8.24e-01 | 8.22e-01 | 4.10e-01 | 0.00e+00 | 3.58e-01 | 5.54e+01 | 5.25e-01 |
| wine quality red | 4.39e-01 | 5.45e-01 | 8.42e-01 | 9.64e-01 | 2.49e-01 | 0.00e+00 | 4.52e-01 | 8.86e+01 | 7.15e-01 |
| wine quality white | 4.28e-01 | 4.99e-01 | 8.15e-01 | 9.54e-01 | 2.52e-01 | 0.00e+00 | 4.48e-01 | 1.88e+02 | 2.33e-01 |
| california | 0.00e+00 | 0.00e+00 | 6.14e-01 | 8.99e-01 | 2.52e-01 | 0.00e+00 | 4.34e-01 | 1.57e+02 | 2.52e-01 |
| bean | 0.00e+00 | 0.00e+00 | 3.32e-01 | 7.50e-01 | 0.00e+00 | 8.06e-01 | 4.03e-01 | 0.00e+00 | 0.00e+00 |
| tictactoe | 1.12e+00 | 1.96e+00 | 8.04e-01 | 6.65e-01 | 0.00e+00 | 7.49e-01 | 3.20e-01 | 0.00e+00 | 0.00e+00 |
| congress | 1.55e+00 | 2.40e+00 | 8.82e-01 | 9.07e-01 | 0.00e+00 | 9.39e-01 | 3.40e-01 | 0.00e+00 | 0.00e+00 |
| car | 5.62e-01 | 1.07e+00 | 4.95e-01 | 5.12e-01 | 0.00e+00 | 7.61e-01 | 3.36e-01 | 0.00e+00 | 0.00e+00 |

Table 10: Runtime results for UnmaskingTrees on benchmark of 27 datasets.

| Dataset | # Samples | # Features | Imputation time (s) | Generation time (s) |
|---|---|---|---|---|
| iris | 150 | 4 | 5.31 | 10.72 |
| wine | 178 | 13 | 26.76 | 49.06 |
| parkinsons | 195 | 22 | 58.98 | 105.27 |
| climate model crashes | 540 | 18 | 103.73 | 207.70 |
| concrete compression | 1030 | 8 | 47.19 | 123.10 |
| yacht hydrodynamics | 308 | 6 | 8.89 | 22.36 |
| airfoil self noise | 1503 | 5 | 29.91 | 92.74 |
| connectionist bench sonar | 208 | 60 | 440.60 | 685.94 |
| ionosphere | 351 | 33 | 201.81 | 362.05 |
| qsar biodegradation | 1055 | 41 | 560.58 | 909.87 |
| seeds | 210 | 7 | 14.94 | 27.65 |
| glass | 214 | 9 | 17.12 | 33.78 |
| ecoli | 336 | 7 | 14.69 | 32.56 |
| yeast | 1484 | 8 | 62.96 | 150.93 |
| libras | 360 | 90 | 1975.78 | 2986.78 |
| planning relax | 182 | 12 | 25.25 | 46.18 |
| blood transfusion | 748 | 4 | 13.16 | 37.24 |
| breast cancer diagnostic | 569 | 30 | 279.33 | 495.71 |
| connectionist bench vowel | 990 | 10 | 79.85 | 179.48 |
| concrete slump | 103 | 7 | 9.74 | 16.49 |
| wine quality red | 1599 | 10 | 106.23 | 263.92 |
| wine quality white | 4898 | 11 | 357.68 | 890.90 |
| california | 20640 | 8 | 968.14 | 2754.75 |
| bean | 13611 | 16 | 1929.16 | 4345.50 |
| tictactoe | 958 | 9 | 25.85 | 51.77 |
| congress | 435 | 16 | 30.41 | 52.56 |
| car | 1728 | 6 | 30.18 | 60.38 |

(A)                                    (B)

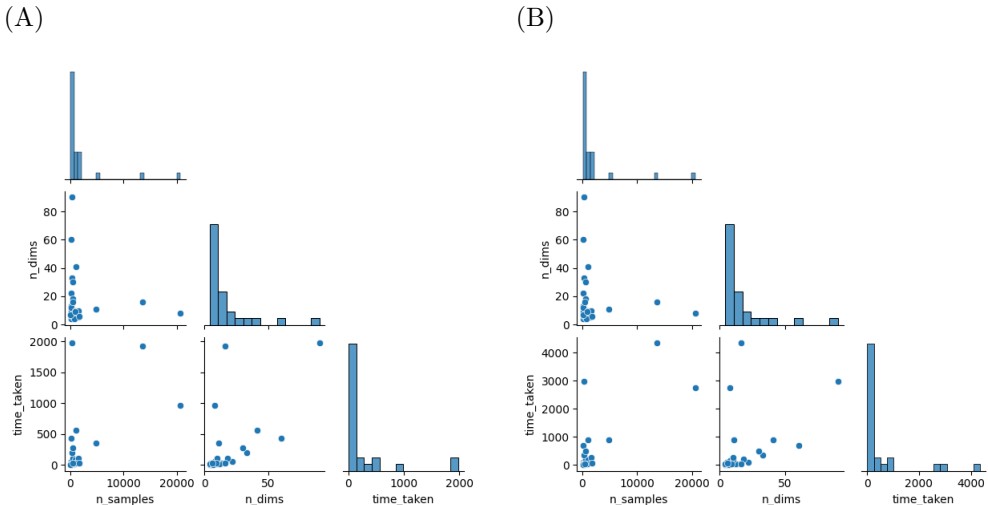

Figure 5: Runtime in seconds compared to number of features and number of samples, for imputation (A) and generation (B) tasks.

