# OpenReview forum: "Unmasking Trees for Tabular Data"
_TMLR — Accepted by TMLR_

### Review · Reviewer_L1KV · 2025-03-02

**Summary Of Contributions:**

The authors propose a new continuous value estimation method, BaltoBot, with application in missing value imputation. The BaltoBot is a new alternative of a regression tree, predicting the split of the node with KDI estimation and Xgboost.

**Audience:**

Yes

**Broader Impact Concerns:**

No need

**Claims And Evidence:**

Yes

**Requested Changes:**

1. Give more motivation or mathematical foundation of the BaltoBot method
2. Compare with TabMT

**Strengths And Weaknesses:**

Strengths:
1. Multimodal prediction of continuous values are important. The authors aims to solve an important problem
2. The authors provide an interesting solution with the combination of KDI and Xgboost.

Weaknesses:
1. The description of the KDI part needs to add more details. Why do you choose KDI? Is it a mathematically justified approach to split the space by KDI and use Xgboost to predict the result from KDI? What's the motivation behind this?
2. The authors menetion TabMT in the paper multiple times. It is worthwhile to compare with the TabMT method in the experiment.

---

> ### Author Response · Authors · 2025-05-31
> **Re: Review by L1KV**
>
> Thank you for your time and attention.
>
> > Re: Summary Of Contributions
>
> We would like to clarify a few points about our contributions. First, BaltoBot is not our sole contribution. We also propose and show that, given data with missingness, it is advantageous to combine autoregression with GBDTs rather than (as recently proposed by ForestDiffusion, AISTATS2024) to combine diffusion with GBDTs. Second, our approach is not solely applicable to imputation, even though it is motivated by the struggles of diffusion in this setting. Notably, we also have SotA results on the benchmark for generation given data with missingness. Third, BaltoBot is not exactly a "continuous value estimation method" nor an "alternative to regression trees". Regression trees try to predict the conditional expectation. BaltoBot is trying to do something different: to predict the conditional distribution, from which one can then generate samples. This makes it an alternative to quantile estimation and univariate diffusion (eg Treeffuser, NeurIPS2024) methods.
>
> > Re: "The description of the KDI part needs to add more details. Why do you choose KDI?"
>
> Thank you for pointing this out this shortfall in our manuscript. The empirical justification is that it works better (see Table 2 ablation experiment), but the intuition behind it is that tabular data often has features which are irregular or highly skewed. KDI was previously shown (McCarter, 2023) to be better at discretizing such highly-skewed features than KMeans-based discretization. KMeans discretization is based on mixture modeling with equal-sized Gaussian components; in constrast, KDI is a 1d density-based clustering method looks for local minima after applying a smoothing transformation. The latter is especially useful for features where there is one big low-variance cluster and then a small number of outliers: only KDI will typically create a bin for those outliers. We revised the manuscript, adding a version of the above to the final paragraph of Section 2.2.
>
> > Re: "Is it a mathematically justified approach to split the space by KDI and use Xgboost to predict the result from KDI? What's the motivation behind this?"
>
> Hierarchical binning addresses the key problem with flat binning, that it destroys proximity information among bins. As we said in Section 2.2: "The simplest solution is to quantize continuous features into bins... Yet this not only destroys information within bins due to rounding, it also destroys information about the proximity among the ordered bins. Thus, it forces us to choose between a small number of quantization bins, yielding low resolution; or to choose a large number of bins, risking catastrophic errors due to overfitting and/or clumping of generated samples due to poor calibration."
>
> The concept of hierarchical partitioning is of course a workhorse in math and computing (eg binary search trees). Indeed, it is how integers are represented in binary format, and its advantages are why place-value numeral systems are preferred to unary "one-hot" representation. And there's also an interesting connection to diffusion. As we note in the Discussion -- see especially the (Dieleman, 2024) reference -- "diffusion is autoregression in frequency space, progressing from low frequencies to high frequencies". On each feature, BaltoBot also progresses from low-frequencies to high-frequencies, allotting more training computation to low-frequencies.
>
> > Re: "It is worthwhile to compare with the TabMT method in the experiment."
>
> There are multiple reasons we do not. First, as noted in our footnote in page 7, "We do not add TabMT (Gulati and Roysdon, 2024) and TabPFGen (Ma et al., 2024) to the benchmark because no code was provided." Speaking frankly, we are frustrated that a paper published at NeurIPS2024 comes without available code for reproducibility. In our opinion, despite its acceptance into NeurIPS, we think one must not be required to include such irreproducible papers as baselines. Second, in our case, doing so would be prohibitive, given that TabMT would clearly not be runnable on our available hardware (a 2015 iMac). Third, the fact that TabMT requires expensive GPU training, unlike ForestDiffusion and UnmaskingTrees, is an important motivation for our work, as we are focusing on scenarios where computing resources are limited. Fourth, a central focus of this paper is to compare diffusion vs autoregression on tabular data. Our comparison with ForestDiffusion is better for directly evaluating this on entire datasets, because it controls for modeling approach (ie XGBoost vs Transformer), revealing the effect of self-supervision approach (ie diffusion/flow-matching vs autoregression); and our comparison with Treeffuser is better for directly evaluating this on 1d probabilistic predictions.

---

> > ### Comment · Reviewer_L1KV · 2025-06-03
> > **Thanks for the response**
> >
> > Thanks for the detailed response. I do not have further questions.

---

### Review · Reviewer_ogxt · 2025-04-08

**Summary Of Contributions:**

The submission, titled "Unmasking Trees for Tabular Data", presents a new method addressing challenges in tabular data imputation and generation, with these key contributions:

1. Introduction of UnmaskingTrees:

- A new approach leveraging gradient-boosted decision trees to incrementally "unmask" individual features in tabular datasets.

- Achieves good performance in imputation and synthetic data generation, particularly when training data contains missing values.


2. Development of BaltoBot:

- A tabular probabilistic prediction method that fits a balanced tree of boosted tree classifiers.

- Unlike traditional methods, BaltoBot does not assume a parametric form for the conditional distribution, making it suitable for handling features with multimodal distributions.

- Offers fast sampling, closed-form density estimation, and a flexible approach to discrete variables, distinguishing it from contemporary diffusion methods.


3. Meta-Algorithm Approach:

- The authors consider both UnmaskingTrees and BaltoBot as meta-algorithms.

- They illustrate the application of in-context learning-based generative modeling with TabPFN.


4. Practical Implementation and Benchmarking:

- The code for UnmaskingTrees is available at GitHub, facilitating broader use within the machine learning community.

- Rigorous benchmarking against 27 tabular datasets, demonstrating competitive performance compared to existing imputation methods, such as MissForest and MICE-Forest, particularly on diversity and accuracy of imputations.


5. Addressing Missingness in Tabular Data:

- The method provides new insights into the handling of missing values, showcasing superior imputation capabilities over several traditional and contemporary methods.

- Recognizes limitations in current literature, suggesting that traditional statistical methods like MissForest still hold considerable ground in specific scenarios.

**Audience:**

Yes

**Broader Impact Concerns:**

n.a.

**Claims And Evidence:**

No

**Requested Changes:**

- Regarding the notation, in instances such as "Consider a dataset with N examples," would it be more appropriate to utilize the term "samples" for consistency within the machine learning literature?

- Concerning Algorithm 1, the presence of three nested iterative structures raises questions about the computational complexity when applied to large-scale datasets. Could the authors provide an analysis of the running time and discuss the feasibility of execution on such datasets?

- In Algorithm 2, could the authors elaborate on the criteria and methodology employed for the selection of the hyperparameter H?

- Within Section 2.3, "Computational complexity," the current comparison is limited to ForestDiffusion. To provide a more comprehensive analysis, would the authors consider including a comparative evaluation against an additional two to three relevant baseline methods?

- In Section 2.4, addressing the handling of missing values (NaNs), could the authors comment on the potential efficacy and implications of replacing NaNs with a specific numerical value, such as -1?

- As noted in the "Weaknesses" section of the review, a comprehensive hyperparameter tuning procedure for all evaluated methods is anticipated in Section 3 to ensure a fair and rigorous comparison.

- In Section 3.1, the inclusion of quantitative results, potentially in tabular form, would significantly enhance the clarity and facilitate a direct comparison of the proposed method's performance.

- With respect to Section 3.2, the observation that "MissForest achieves the top rank on 4 out of 9 metrics" suggests a performance level comparable to the proposed method. To statistically validate any perceived superiority, the application of a Nemenyi post-hoc test, as referenced in [1], might be beneficial.

- Regarding Table 3, given that Tabsyn [2] currently represents the state-of-the-art, its inclusion in the comparative analysis would provide a more accurate assessment of the proposed method's position within the field.

- In Table 4, the results indicate that Forest-Flow exhibits the best performance. Could the authors provide a more detailed explanation of the underlying reasons for this observation?

- Within Section 3.3, concerning the "Wave dataset" experiments, could the authors justify the limited comparison to only Treeffuser and consider including results from other relevant methods for a more thorough evaluation?

- For Figure 4, enhancing the visual comparison by including the results of a couple of additional relevant methods alongside Treeffuser and the proposed approaches would provide a more comprehensive understanding of the performance landscape.

- Regarding the manuscript structure, specifically Section 5, it is suggested that the "Related Work" section be moved to the beginning of the paper and that the "Discussion" section be integrated with the "Results" section to improve the logical flow and clarity for the reader.

- Given the results presented, where the proposed method does not consistently outperform other approaches, it is recommended that the authors moderate the claim of achieving "state-of-the-art" performance.

---
[1] Janez Demšar. Statistical comparisons of classifiers over multiple data sets. Journal of Machine Learning Research, 7:1–30, 2006. ISSN 15337928.

[2] Zhang, H., Zhang, J., Shen, Z., Srinivasan, B., Qin, X., Faloutsos, C., Rangwala, H., & Karypis, G. (2024). Mixed-Type Tabular Data Synthesis with Score-based Diffusion in Latent Space. In The Twelfth International Conference on Learning Representations.

**Strengths And Weaknesses:**

**Strengths**

1. New Approach:

- The introduction of UnmaskingTrees presents a unique method that enhances traditional tabular data handling by focusing on feature unmasking, showcasing advancements in imputation and generation tasks.

2. Performance:

- Empirical results demonstrate that the proposed method achieves decent performance on various benchmarks. This positions it as a reliable solution for both imputation (handling missing values) and generation (creating synthetic data).

3. BaltoBot's Flexibility:

- The development of BaltoBot provides a robust alternative to conventional imputation methods by avoiding parametric assumptions, allowing it to accommodate complex, multimodal distributions in tabular data.

4. Practical Code Availability:

- The authors have contributed to the community by making their implementation available via a GitHub repository, facilitating further research and practical applications of their methods.


---
**Weaknesses**

1. Existing Literature Limitations:

- The paper acknowledges that traditional methods, such as MissForest, have not been completely surpassed in all metrics, particularly those based on Wasserstein distances. This raises questions about whether the proposed method can universally outperform all existing approaches?


2. Generalizability Concerns:

- While the performance on benchmark datasets is promising, it would benefit the authors to include more diverse datasets to test the generalizability of their methods. Specific edge cases or datasets with unique characteristics may reveal shortcomings not currently addressed.

- The paper would gain credibility by extending the benchmark to include medium- and large-sized datasets, as the current evaluation is limited to smaller ones.


3. Hyperparameter Tuning:

- The submission mentions optimized hyperparameter spaces for XGBoost but lacks a thorough exploration of how sensitive the performance is to these hyperparameters. Including a detailed analysis of hyperparameter tuning would strengthen the robustness of the methodology.

- For a fair comparison, the authors should perform hyperparameter tuning for each evaluated method, rather than using default settings.


4. Clarification on Limitations:

- Further elaboration on the limitations of the UnmaskingTrees approach, particularly in terms of scalability for larger datasets, would provide a more balanced view of its applicability in real-world scenarios.


5. Future Work Directions:

- The paper would benefit from a more explicit discussion on future research paths. Specifying potential enhancements or explorations that could be made to address current limitations would provide insight into the broader impact of this work.

---

> ### Author Response · Authors · 2025-05-31
> **Re: Review by ogxt (part 1)**
>
> Thank you for your thorough consideration and feedback.
>
> Summary and Strengths
>
> We agree with your summary and appreciate your recognition of the strengths of our paper.
>
> Weaknesses
>
> > (W1) The paper acknowledges that traditional methods, such as MissForest, have not been completely surpassed in all metrics, particularly those based on Wasserstein distances. This raises questions about whether the proposed method can universally outperform all existing approaches?
>
> We do not claim that it does, and indeed say in Limitations, "It remains to be seen whether a single method can be developed which wins on all scenarios and metrics." Given that TMLR's acceptance criteria are "evidence for claims" and "interestingness", we do not believe that universally outperforming all existing approaches is required for acceptance.
>
> > (W2) it would benefit the authors to include more diverse datasets to test the generalizability of their methods
>
> The Jolicouer-Martineau et al (2024)'s benchmark already contains a diverse selection of 27 datasets. More info about this benchmark is given in Table 10 of our Appendix, and in Appendix B1 of their paper. We would also like to gently point out that many (if not most!) tabular data papers choose (ie cherry-pick) their own selection of datasets on which to evaluate their methods. It is comparatively much harder to show improvements on a previously-published benchmark, as we have done.
>
> > (W3) The paper would gain credibility by extending the benchmark to include medium- and large-sized datasets, as the current evaluation is limited to smaller ones.
>
> As shown in Table 10, Jolicouer-Martineau et al (2024)'s benchmark contains datasets with up to 20640 samples (California) and 90 features (Libras). In the Introduction, we have clarified that "we are primarily focused on generation and imputation methods for users with limited data and computing resources". In Limitations, we have clarified with: "we would like to emphasize that our proposed approach is aimed at and evaluated on smaller-sized tabular datasets. It is also evaluated via “out-of-the-box” performance, being aimed at users lacking the resources for large deep learning models or hyperparameter optimization. For users with access to larger tabular datasets and more extensive computing resources, recent deep learning methods like Tabsyn (Zhang et al., 2024) would be expected to perform better."
>
> > (W4) The submission mentions optimized hyperparameter spaces for XGBoost but lacks a thorough exploration of how sensitive the performance is to these hyperparameters. Including a detailed analysis of hyperparameter tuning would strengthen the robustness of the methodology.
>
> On the contrary, we do not optimize XGBoost hyperparameters at all for the (Jolicouer-Martineau et al, 2024) benchmark, instead using its default hyperparameters.
>
> > (W5) For a fair comparison, the authors should perform hyperparameter tuning for each evaluated method, rather than using default settings.
>
> We use the (Jolicouer-Martineau et al, 2024) benchmark as-is, which measures the "out-of-the-box" performance of all methods, including our own, with default hyperparameters. So these results are completely fair. We note that using default hyperparameters is common in the small tabular datasets literature. Besides (Jolicouer-Martineau et al, 2024), see these recent examples:
>
> Shi, X., Mueller, J., Erickson, N., Li, M., & Smola, A. (2021). Benchmarking Multimodal AutoML for Tabular Data with Text Fields. In Thirty-fifth Conference on NeurIPS Datasets and Benchmarks Track.
>
> Gorishniy, Y., Rubachev, I., Khrulkov, V., & Babenko, A. (2021). Revisiting deep learning models for tabular data. Advances in neural information processing systems, 34, 18932-18943.
>
> Hamad, F., Nakamura-Sakai, S., Obitayo, S., & Potluru, V. (2023, November). A supervised generative optimization approach for tabular data. In Proceedings of the Fourth ACM International Conference on AI in Finance (pp. 10-18).
>
> Wang, A. X., Chukova, S. S., Simpson, C. R., & Nguyen, B. P. (2024). Challenges and opportunities of generative models on tabular data. Applied Soft Computing, 112223.
>
> Cresswell, J. C., & Kim, T. (2024). Scaling Up Diffusion and Flow-based XGBoost Models. arXiv preprint arXiv:2408.16046.
>
> > (W6) Further elaboration on the limitations of the UnmaskingTrees approach, particularly in terms of scalability for larger datasets
>
> Done.

---

> > ### Author Response · Authors · 2025-05-31
> > **Re: Review by ogxt (part 2)**
> >
> > Weaknesses (continued)
> >
> > > (W7) The paper would benefit from a more explicit discussion on future research paths.
> >
> > We added the following to the Limitations and Future Work section: "Reducing the training complexity with respect to the number of features is a key next step. One possibility would be to use an optimized, rather than random, selection of feature orderings at training time (Shih et al., 2022). Another possibility would be to use multi-output trees to train a single XGBoost model for all BaltoBot tree nodes and all features, similarly to a recently-proposed approach for speeding up ForestDiffusion (Cresswell and Kim, 2024). In addition to improving scalability, BaltoBot’s core idea of BaltoBot’s binary partitioning could be combined with deep learning approaches. One might equip a neural network with a hierarchical softmax head (Morin and Bengio, 2005) for modeling continuous outputs without losing proximity information among bins."
> >
> > Requested changes
> >
> > > in instances such as "Consider a dataset with N examples," would it be more appropriate to utilize the term "samples"
> >
> > Done.
> >
> > > (R1) Could the authors provide an analysis of the running time and discuss the feasibility of execution on such datasets?
> >
> > In addition to the big-O analysis in "Section 2.3 Computational complexity", Appendix B contains the following additional results, showing large runtime improvements over ForestDiffusion and ForestFlow on the largest datasets in their benchmark:
> >
> > "Timing results are in Table 10, and depicted in Figure 5. Our method is relatively efficient at both imputation and generation, compared to diffusion / flow-based methods. The datasets on which we are slowest for imputation are Libras (1976 seconds, N = 360, D = 90) and Bean (1929 seconds, N = 13611, D = 16), on our ancient 2015 iMac with 16Gb RAM. On Libras, ForestVP imputation took 12439 seconds (without RePaint) and 14715 seconds (with RePaint); on Bean, ForestVP took 898 seconds (without RePaint) and 1318 seconds (with RePaint), on their cluster of 10-20 CPUs with 64-256Gb of RAM. The datasets on which we are slowest for generation are also Libras (2987 seconds) and Bean (4346 seconds). On Libras, ForestFlow generation took 9481 seconds and ForestVP took 9042 seconds; on Bean, ForestFlow took 869 seconds and ForestVP took 947 seconds, once again on their much more powerful computing cluster."
> >
> >
> > > (R2) In Algorithm 2, could the authors elaborate on the criteria and methodology employed for the selection of the hyperparameter H?
> >
> > Please see Section 3: "Results were obtained always using our method's default hyperparameters: BaltoBot tree height of 4, and duplication factor K=50. These hyperparameter values were tuned on the Two Moons and Iris case studies, then applied without further tuning to the remaining experiments." In practice, at least for the Two Moons and Iris datasets, making the H and K bigger made the method slower without providing noticeably better quality results.
> >
> > > (R3) a comprehensive hyperparameter tuning procedure for all evaluated methods is anticipated in Section 3 to ensure a fair and rigorous comparison
> >
> > Please see our response above to (W5).
> >
> > > (R4) To provide a more comprehensive analysis, would the authors consider including a comparative evaluation against an additional two to three relevant baseline methods?
> >
> > We have added comparisons with MissForest and MICE-Forest.
> >
> > > (R5) In Section 2.4, addressing the handling of missing values (NaNs), could the authors comment on the potential efficacy and implications of replacing NaNs with a specific numerical value, such as -1?
> >
> > Because TabPFN performs nonlinear feature preprocessing, the effect of such an approach could lead to unexpected behavior, depending on the distribution of observed values.
> >
> > > (R6) In Section 3.1, the inclusion of quantitative results, potentially in tabular form, would significantly enhance the clarity and facilitate a direct comparison of the proposed method's performance.
> >
> > We follow (Jolicouer-Martineau et al, 2024) in depicting the results visually. We also note that aggregated metrics (eg those used in Section 3.2) do not always reflect rare-but-bad mistakes, such as ForestDiffusion's spurious species imputations in Figure 2.
> >
> > > (R7) With respect to Section 3.2, the observation that "MissForest achieves the top rank on 4 out of 9 metrics" suggests a performance level comparable to the proposed method.
> >
> > As described in the notes for our revised manuscript, we now refrain from claiming our approach is state-of-the-art on imputation.

---

> > > ### Author Response · Authors · 2025-05-31
> > > **Re: Review by ogxt (part 3)**
> > >
> > > > (R8) Regarding Table 3, given that Tabsyn [2] currently represents the state-of-the-art, its inclusion in the comparative analysis would provide a more accurate assessment of the proposed method's position within the field.
> > >
> > > Based on our understanding, Tabsyn is more appropriate for users with more data and more compute than our current work is focused on. We added the following to Limitations: "For users with access to larger tabular datasets and more extensive computing resources, recent deep learning methods like Tabsyn (Zhang et al., 2024) would be expected to perform better."
> > >
> > > > (R9) In Table 4, the results indicate that Forest-Flow exhibits the best performance. Could the authors provide a more detailed explanation of the underlying reasons for this observation?
> > >
> > > The second paragraph of Discussion and Related Work concludes with the following: "On the other hand, the advantages of diffusion modeling (no quantization error, holistic generation, needing only an estimated score function rather than well-calibrated conditional distributions) give it superiority when these problems can be avoided."
> > >
> > > > (R10) Within Section 3.3, concerning the "Wave dataset" experiments, could the authors justify the limited comparison to only Treeffuser and consider including results from other relevant methods for a more thorough evaluation?
> > >
> > > The "Wave dataset" was introduced by Treeffuser for illustrative purposes, to show that univariate diffusion can predict heteroskedastic, multimodal conditional distributions. We use it similarly, to illustrate the capability of BaltoBot to match diffusion in such settings.
> > >
> > > > (R11) For Figure 4, enhancing the visual comparison by including the results of a couple of additional relevant methods alongside Treeffuser and the proposed approaches would provide a more comprehensive understanding of the performance landscape.
> > >
> > > Our primary claim in the paper is that, equipped with BaltoBot as a subroutine, autoregression can outperform diffusion for tabular imputation. We are not primarily evaluating BaltoBot as a standalone method, and to the extent that we do evaluate it, we are primarily claiming that it has benefits vs diffusion. A full, careful evaluation of probabilistic forecasting approaches is complex and outside the scope of our paper. Note that Treeffuser is a 1d version of ForestDiffusion evaluated on probabilistic forecasting tasks; it appeared several months after ForestDiffusion and was accepted into NeurIPS2024.
> > >
> > > > (R12) Regarding the manuscript structure, specifically Section 5, it is suggested that the "Related Work" section be moved to the beginning of the paper and that the "Discussion" section be integrated with the "Results" section to improve the logical flow and clarity for the reader.
> > >
> > > Thanks for the suggestion; we will potentially consider this in any future revisions.
> > >
> > > > (R13) Given the results presented, where the proposed method does not consistently outperform other approaches, it is recommended that the authors moderate the claim of achieving "state-of-the-art" performance.
> > >
> > > Upon reflection, we agree and have updated our manuscript accordingly.

---

> > > > ### Comment · Reviewer_ogxt · 2025-06-04
> > > >
> > > > Thank you very much for your detailed responses; I truly appreciate the thoroughness. I have a few questions and observations that I'd be grateful if you could clarify.
> > > >
> > > > 1.  I've been reviewing the benchmark by Jolicouer-Martineau et al. (2024) and noticed that the reported results for the evaluated methods in your manuscript appear to differ. For my better understanding, could you kindly elaborate on the reasons behind these differences and perhaps explain your methodology or any specific considerations that led to these varied outcomes?
> > > >
> > > > 2.  My understanding and experience with methods like GaussianCopula, TVAE, CTGAN or TabDDPM suggest that they generally perform well across various dataset sizes, including small, medium, and large. Given your stated focus on "smaller-sized tabular datasets," I'm wondering if a comparison with these broader-scoped methods is the most appropriate. Perhaps you could clarify the rationale for including them, or consider whether other benchmarks might offer a more direct comparison given your dataset size focus.
> > > >
> > > > 3.  Building on my previous point regarding the comparison methods (GaussianCopula, TVAE, TabDDPM, etc., as mentioned in point 2), this is precisely why I suggested including TabSyn ([https://arxiv.org/abs/2310.09656](https://arxiv.org/abs/2310.09656)) for comparison. It seems it might align well with the context of your existing comparative analysis.
> > > >
> > > > 4.  I tend to lean towards ensuring the fairest possible comparisons. Therefore, I would still gently suggest that a comprehensive hyper-parameter search for all methods, rather than relying on default or selected parameters, would significantly strengthen the comparative analysis in your study.

---

> > > > > ### Author Response · Authors · 2025-06-05
> > > > > **Re: Official Comment by Reviewer ogxt**
> > > > >
> > > > > 1. The reported numbers in the main text of both ours and Jolicouer-Martineau et al. (2024)'s are average ranks for each metric. What this means (for a particular metric) is that for each of the 27 datasets, one computes the ranks of all methods relative to each other. Then the average ranks are computed as the average over the aforementioned 27 datasets. As one introduces new methods, these numbers will move upward, because the average of the average ranks is (the number of methods + 1)/2.
> > > > >
> > > > > The raw values that we used for other methods were simply those which were provided at https://github.com/SamsungSAILMontreal/ForestDiffusion/tree/main/Results (Jolicouer-Martineau et al's code repository), then appending our raw values (reported in Tables 7, 8, and 9). And that URL contains also contains the scripts for computing average rank.
> > > > >
> > > > > 2. In the case of imputation, MissForest and MICE-Forest are especially geared towards small datasets, and are the methods used by the vast majority of statisticians and small-N-focused practitioners. In the case of generation, we think the comparison methods, which are the same ones used by Jolicouer-Martineau et al (2024), have not become too outdated since then, especially for small-N settings. As far as we can tell, the vast majority of recent papers for both imputation and generation have evaluated on benchmarks primarily containing datasets with thousands rather than hundreds of samples.
> > > > >
> > > > > 3. We agree that this would be interesting analysis. But: (1) We think it would be even more appropriate in the context of our planned Future Work, on combining binary partitioning-based prediction heads with deep networks. Then one could ablate TabSyn with a version of it that uses autoregression and binary partitioning. (2) Our experiments are currently limited by computing resources, which as stated previously consists of a 2015 iMac. We are hopeful to obtain more resources soon, but evaluating TabSyn right now is not straightforward.
> > > > >
> > > > > 4. We do appreciate the fact that, for large datasets, reporting untuned hyperparameter performance would be unrepresentative of real-world usage. But: (1) "Out-of-the-box" performance is representative of real-world usage for those with data and/or computing constraints. Indeed, for tuned hyperparameter performance, the Jolicouer-Martineau et al (2024) benchmark probably does not contain appropriately-sized datasets.  So we think that the current results provide sufficient evidence for our (revised) claims, and will be interesting to the portion of the audience with such constraints, thus meeting the TMLR criteria. (2) Whether we were to use the Jolicouer-Martineau et al (2024) benchmark or create a new benchmark for tuned hyperparameter performance, it is currently out of reach of our available computing. Note that imputation evaluation involves multiple experiments (ie induced missingness patterns) for each dataset, and also multiple imputation runs for each experiment to measure diversity. Optimizing hyperparameters for previous works (especially GAIN which often requires thousands of epochs of GAN training) is currently infeasible for us right now. (3) Hyperparameter optimization for multiple imputation methods is itself tricky: the proper choice of cross-validation metric is not straightforward because for we do not necessarily want to impute the closest-to-groundtruth value, but instead want to balance accuracy and diversity. (If we're being completely honest, while Jolicouer-Martineau et al's creation of a standardized benchmark was a huge step in the right direction, what imputation needs most right now are better benchmarks, not more methods. This also is "future work".)

---

> > > > > > ### Comment · Reviewer_ogxt · 2025-06-07
> > > > > >
> > > > > > Thanks for your responses.

---

### Review · Reviewer_cvRQ · 2025-05-16

**Summary Of Contributions:**

- A permutation-based autoregressive framework, _UnmaskingTrees_, with gradient-boosted trees for tabular data imputation and generation.
  - A novel approach, _BaltoBot_, to model conditional distributions via a balanced meta-tree of binary classifiers that allows fast sampling and closed-form density estimates.
  - Extensive numerical experiments that show that the two methods combined achieve SOTA performance on generation with missing data, are comparable to SOTA on imputation with missing data, and are competitive on vanilla generation.
  - A python package. (Great choice of emojis!)

**Audience:**

Yes

**Broader Impact Concerns:**

N/A.

**Claims And Evidence:**

Yes

**Requested Changes:**

These suggestions are not critical, but they would help clarify the presentation and thus improve the paper.

Here are some passages that I found confusing, and would appreciate if they were clarified/reviewed:

  - In Section 2, I think this could be more effectively framed as a single-method paper. I found that introducing two methods broke the flow and added confusion. My point: in UnmaskingTrees, wouldn’t it make more sense to use BaltoBot for everything that isn’t categorical? BaltoBot works for both continuous and discrete data, so the use of XGBClassifier only makes sense for categorical inputs. That said, I don’t think it’s necessary to restructure the entire exposition.

  - In 2.1, _"UnmaskingTrees combines the gradient-boosted trees of ForestDiffusion.."_--isn't that just XGBoost? Aside from using XGBoost, I find no similarities between the proposed autoregressive approach and the diffusion setting in Jolicoeur-Martineau et al. (2024b), so it’s unclear why the comparison is being emphasized.

  - In 2.2, _"A key problem when autoregressively generating continuous data is that a regression model will attempt to predict the mean of a conditional distribution, whereas we would like it to sample from the possibly-multimodal conditional distribution."_ I understand the point, but I find mentioning "autoregressive modeling", "continuous responses", and "regression model" could be confusing here. The core idea, I believe, is this: we want to use trees to sample from a univariate distribution. To do that, we frame the problem as a classification task over a partition of the variable’s support. The key question is then: how are these intervals defined? (Again, this problem doesn't apply only to continuous variables, but to any non-categorical data, such as count data (as shown in 3.3--4).)

  - In 2.2, worth add a sentence at the end of this section that explains that this can be naturally applied to a response with discrete outcome, as validated in the experiments in sections 3.3 and 3.4.

  - In Table 1--3, why is ForestDiffusion the main benchmark for all tasks (I'm referring to the boldface blue color)?

Some questions:

  - Do you have a sense of why MissForest outperforms the proposed approach on some metrics, while UnmaskingTrees performs much better on others? In which scenarios would you prefer one method over the other?

  - In its essence, BaltoBot estimates a conditional density with a mixture of uniforms over a partition of the variabe's support. The height $H$ of the ``meta-tree'' determines the granularity of this partition, and so seems to be a key parameter. Could you discuss how to choose $H$, and more generally, whether you have a sense of how it affects performance?

**Strengths And Weaknesses:**

### Strengths
  - Autoregressive modeling with trees is a novel and interesting idea, and it's shown to be effective here.
  - The paper includes extensive empirical validation.

### Weaknesses
  - The exposition in the methods section could be clearer.

---

> ### Author Response · Authors · 2025-06-01
> **Re: review by cvRQ**
>
> We thank the reviewer for their attention and valuable suggestions.
>
> > In Section 2, I think this could be more effectively framed as a single-method paper. I found that introducing two methods broke the flow and added confusion. My point: in UnmaskingTrees, wouldn’t it make more sense to use BaltoBot for everything that isn’t categorical? BaltoBot works for both continuous and discrete data, so the use of XGBClassifier only makes sense for categorical inputs.
>
> Thanks for pointing this out. We have in fact used BaltoBot for both continuous and discrete data, only using XGBClassifier for categorical inputs. We have updated Section 2 and Algorithm 1 to reflect this.
>
> > In 2.1, "UnmaskingTrees combines the gradient-boosted trees of ForestDiffusion.."--isn't that just XGBoost? Aside from using XGBoost, I find no similarities between the proposed autoregressive approach and the diffusion setting in Jolicoeur-Martineau et al. (2024b), so it’s unclear why the comparison is being emphasized.
>
> Good point -- we've updated this sentence to reflect this.
>
> > In 2.2, "A key problem when autoregressively generating continuous data is that a regression model will attempt to predict the mean of a conditional distribution, whereas we would like it to sample from the possibly-multimodal conditional distribution." I understand the point, but I find mentioning "autoregressive modeling", "continuous responses", and "regression model" could be confusing here.
>
> What we are trying to point at is that switching from diffusion to autoregression introduces a complication that we need to address. With diffusion you only need to predict the conditional mean (the score function in Tweedie's formula), but switching to autoregression means we must now explicitly predict the conditional distribution.
>
>  > In 2.2, worth add a sentence at the end of this section that explains that this can be naturally applied to a response with discrete outcome, as validated in the experiments in sections 3.3 and 3.4.
>
> We revised Section 2.2 to say, "Second, our singleton-bin technique allows us to adaptively generate discrete and mixed-type variables, if the discrete outcome is frequent relative to the total number of training samples and to the size of the meta-tree."
>
> > In Table 1--3, why is ForestDiffusion the main benchmark for all tasks (I'm referring to the boldface blue color)?
>
> One of the goals of this paper (noted in Introduction and Discussion) is to argue for the applicability of autoregression for tabular data with missingness. These highlights make it easier for the reader to see which modeling approach (diffusion vs flow-matching vs autoregression) worked best. The other methods' performances confound differences in modeling approach with other differences (e.g. using random forests or neural networks).
>
> > Do you have a sense of why MissForest outperforms the proposed approach on some metrics, while UnmaskingTrees performs much better on others? In which scenarios would you prefer one method over the other?
>
> MissForest's success on the Wasserstein distance metrics highlights its strength: the distribution of the imputed data is close to the distribution of the training data (and the distribution of the test data). While UnmaskingTrees rarely produces atypical samples, the distribution of the imputed samples is not always close. The way this can happen is that the model for a particular feature can easily learn a miscalibrated conditional distribution, so samples from it are not fully representative of the training data. Even when this happens, the imputed data will not tend to be atypical, but the distribution will still be distorted. (MissForest and MICE-Forest, like ForestDiffusion, require predicting the conditional mean, not the conditional distribution.)
>
> > In its essence, BaltoBot estimates a conditional density with a mixture of uniforms over a partition of the variabe's support. The height H of the "meta-tree'' determines the granularity of this partition, and so seems to be a key parameter. Could you discuss how to choose H, and more generally, whether you have a sense of how it affects performance?
>
> Unlike with standard flat quantization where having a large number of bins can cause one to make catastrophically wrong predictions, BaltoBot "knows" proximities among meta-tree leaves. This means that making H bigger tends not to cause major errors, so it's better to err on the side of larger H rather than small H. The main drawbacks with increasing H are that (1) training takes longer, and (2) imputations and generations are less diverse. On Two Moons and Iris, we increased H until we saw that the resulting plots stopped showing visible improvement. We probably could have increased H further, but it would've made the other experiments more time-consuming.

---

> > ### Comment · Reviewer_cvRQ · 2025-06-04
> >
> > Thank you for clarifying my questions and including the suggested edits.
> >
> > Based on the discussion above, I would also suggest adding the following to the paper:
> >   - Mention diffusions in the opening sentence of Section 2.2. It makes sense to frame the estimation of the score function in diffusions as a regression problem and making this connection explicit would clarify the exposition.
> >   - Discuss the choice of $H$. This could be addressed in the Limitations and mentioned again in the Results sections.

---

> > > ### Author Response · Authors · 2025-06-05
> > > **Re: Official Comment by Reviewer cvRQ**
> > >
> > > Thanks for these additional suggestions.
> > >
> > > We now begin Section 2.2 withe the following: "In diffusion models, predicting the score function can be framed as a regression problem where the model learns to estimate the conditional mean. However, a key problem when switching to autoregressively generating continuous data is that this regression approach will attempt to predict the mean of a conditional distribution, whereas we would like the model to sample from the possibly-multimodal conditional distribution."
> > >
> > > We've also expanded discussion of H in both Results and Limitations, where we added the following: "Our proposed BaltoBot method would benefit from a more principled method for selection of the meta-tree height H. Unlike with standard flat quantization where having a large number of bins can cause one to make catastrophically wrong predictions, BaltoBot “knows” proximities among meta-tree leaves. This means that making H bigger tends not to cause major errors, so it’s better to err on the side of larger H rather than small H. Still, the main drawbacks with increasing H are that (1) training takes longer, and (2) imputations and generations are less diverse. Deeper theoretical analysis of these trade-offs and a more principled approach for choosing the height would improve the ease-of-use and potentially the performance of our method."

---

> > > > ### Comment · Reviewer_cvRQ · 2025-06-06
> > > >
> > > > Appreciate the revisions—I have no further comments.

---

### Decision · Action_Editor_g6AE · 2025-06-29

**Recommendation:** Accept with minor revision

**Additional Comments:**

It should also be clarified in the abstract that the main claims about performance is on small tabular datasets.

**Audience:**

Yes

**Audience Explanation:**

The main topic of the paper lies with the core topic of TMLR and people working in the intersection between probabilistic machine leanring and missing data.

**Claims And Evidence:**

Yes

**Claims Explanation:**

The three main claims of state-of-the-art performance on imputation and generation with missing data, as well as competitive performance in the vanilla generative setting, are supported by the results on small tabular benchmark datasets

---

> ### Author Response · Authors · 2025-07-03
> **Camera-ready revision**
>
> Thanks to you and to all the reviewers for your attention and careful feedback. The abstract in the camera-ready revision now contains this clarification.